# Provable General Function Class Representation Learning in Multitask Bandits and MDPs

**Rui Lu[1], Andrew Zhao[1], Simon S. Du[2], Gao Huang[1]**
[1]Department of Automation, BNRist, Tsinghua University
[2]Paul G. Allen School of Computer Science and Engineering, University of Washington
{r-lu21,zqc21}@mails.tsinghua.edu.cn
ssdu@cs.washington.com, gaohuang@tsinghua.edu.cn

## Abstract

While multitask representation learning has become a popular approach in reinforcement learning (RL) to boost the sample efficiency, the theoretical understanding of why and how it works is still limited. Most previous analytical works could only assume that the representation function is already known to the agent or from linear function class, since analyzing general function class representation encounters non-trivial technical obstacles such as generalization guarantee, formulation of confidence bound in abstract function space, etc. However, linear-case analysis heavily relies on the particularity of linear function class, while real-world practice usually adopts general non-linear representation functions like neural networks. This significantly reduces its applicability. In this work, we extend the analysis to *general* function class representations. Specifically, we consider an agent playing $M$ contextual bandits (or MDPs) concurrently and extracting a shared representation function $\phi$ from a specific function class $\Phi$ using our proposed Generalized Functional Upper Confidence Bound algorithm (GFUCB). We theoretically validate the benefit of multitask representation learning within general function class for bandits and linear MDP for the first time. Lastly, we conduct experiments to demonstrate the effectiveness of our algorithm with neural net representation.

## 1 Introduction

Recently, reinforcement learning (RL) has achieved many successful applications in games [6, 34], robotics [23], and many other fields. However, due to the large cardinality of state space or action space in real-world problems, the large sample complexity has been a major problem for employing these RL algorithms in reality. A popular method called multitask representation learning tries to tackle this problem by extracting a shared low-dimensional representation function among multiple related tasks, then using a simple function (e.g., linear) on top of this common representation to solve each task[4, 7, 24].

Despite the empirical success for multitask representation learning, particularly in reinforcement learning because of its effectiveness in reducing sample complexity, the theoretical understanding about it is still limited. A march of works[37, 36, 22, 33, 25, 31, 16, 3, 11, 9, 41, 30] give results on function approximation in bandits and RL, which permits a representation. In these frameworks, an agent is considered playing $M$ related tasks concurrently. Each task is a distinct contextual bandit or linear MDP problem [1], and all these $M$ tasks share a common representation $\phi \in \Phi$ where $\Phi = \{\phi : \mathcal{S} \times \mathcal{A} \mapsto \mathbb{R}^k\}$ is representation function class extracting a $k$-dimensional representation

---

[1]Although the name of linear MDP contains term "linear", it actually has infinite degrees of freedom because the representation function $\phi$ could be general non-linear function.

vector from state-action pair. Such representation function can reduce the complexity of problem from a huge space $\mathcal{S} \times \mathcal{A}$ to a simple regression problem in $k$-dimensional space. The value approximation function class is defined by $\mathcal{F} = \mathcal{L} \circ \Phi$, here $\circ$ means composition and $\mathcal{L}$ means linear function, which means the value of any state-action pair $(s, a)$ is linear in its representation $\phi(s, a)$.

However, previous analyses either assume $\Phi$ is linear [40], or assume that the agent already knows the concrete function $\phi$ [17, 21], which equivalently reduces to learning linear weight parameters. This limits their applicability, since general non-linear value estimation is ubiquitous and is the essence for the success of multitask representation learning. For instance, DQN[29] achieves great success by employing a deep network to approximate Q-value function. Also, assuming the agent already knows a good representation function is unrealistic in practice. Therefore, we aim to extend the analysis to *unknown general non-linear* representation functions. This would not only reveal the more essential benefit of multitask representation learning, but also inspire and facilitate future practice.

## 1.1 Our Contribution

The focus of previous works on linear analyses has its own reasons. The particularity of linear function could circumvent many non-trivial obstacles in analysis, which hinders previous work from from being applicable to real world scenarios. For instance, the formulation of confidence set in linear parameter space is simply an ellipsoid, and its update is straightforward via covariance matrix. More importantly, linear function class generically ensures generalization. The analysis [18, 43, 26, 21] only requires the samples to span the whole input space to let the covariance matrix converge, then is able to derive uniform prediction error guarantee for the whole input space. However, generalization issue is much more complicated for general non-linear scenarios.

In summary, our work embraces following contributions, which solves the challenges for previous works and extends the analysis for the role of representation function in more general setting.

**Eliminate the Dependency on Linearity.** Towards general function class analysis, we adopt the idea of confidence set [32, 17]. The algorithm extends the idea of upper confidence bound and maintains a *confidence set* for all the possible value estimation functions. The confidence set contains all the functions whose total empirical error at step $t$ is less than a predetermined bound $\beta_t$. As more seen data reveals more information about the environment, the confidence set will gradually shrink until converge. Therefore, our algorithm and analysis framework is applicable to general function class.

Note that designing $\beta_t$ to achieve low regret for general function class $\Phi$ is non-trivial. We firstly determine the concrete UCB form for general function class $\beta_t(\Phi)$ and propose a straightforward algorithm called Generalized Functional Upper Confidence Bound (or GFUCB in abbreviation) for general non-linear function class approximation. We use Eluder dimension[32] to measure the complexity of the function class $\Phi$ to give an efficient sample complexity that ensures generalization.

**Multihead Function Class.** To derive sharp regret bound for our algorithm and theoretically demonstrate the benefit of multitask representation learning, we firstly introduce multihead function class $\mathcal{F}^{\otimes M}$, which is the key technical contribution of our work. The efficacy of multitask representation learning essentially originates from the shared knowledge and structure among tasks. Hence it is vital and necessary to characterize such relation between multiples tasks that the agent simultaneously learns. However, such structure is absent in previous single task work [39, 32], and it calls for special techniques to analyze the efficiency for learning these correlated functions.

To this end, we introduce multihead function class, namely $\mathcal{F}^{\otimes M}$ in section 4. This abstract function space captures the relation between different task functions, which concatenate the values of $(s, a)$ for all $M$ tasks together as the output. Being more compact by sharing a common backbone $\phi$, function in $\mathcal{F}^{\otimes M}$ requires much fewer samples to learn compared to $M$ independent tasks space $\mathcal{F}^M$. All the tasks contribute to shape a good representation, then feedback to each task for faster convergence. We formally prove that our algorithm enjoys regret bound as $\tilde{O}\left(\sqrt{MT \dim_E(\mathcal{F})(Mk + \log \mathcal{N}(\Phi))}\right)$, where $T$ is the number of steps, $M$ is the number of tasks and $\mathcal{N}(\Phi)$ means the covering number of function space $\Phi$. We also extend the algorithm and analysis to multitask episodic RL with general value approximation under low inherent Bellman error. By simultaneously solving $M$ different but correlated MDP tasks, our method is sample-efficient with regret $\tilde{O}\left(\sqrt{MTH \dim_E(\mathcal{F})(Mk + \log \mathcal{N}(\Phi) + MTH\mathcal{I}^2)}\right)$ where $T$ is the number of episodes, $H$ is planning horizon and $\mathcal{I}$ denotes the inherent Bellman error.

To the best of our knowledge, this is the first provably sample efficient algorithm for general representation function bandits and linear MDP. It is comparable to the most optimal regret bound when $\Phi$ is specialized to linear representation, and is better than the bounds which solve each task independently. This also theoretically explains how multitask representation learning reduces sample complexity. Essentially, the joint training for the shared representation function helps accelerate the convergence of the common backbone by having more samples from all the tasks.

**Empirical Value.** Finally, we conduct experiments to verify our theoretical result. We design a neural network based bandit environment and implement the GFUCB algorithm. Experimental results corroborate the effect of multitask representation learning in boosting sample efficiency in non-linear bandits. For the first time, the efficacy of the general representation algorithm proposed in theoretical analysis is validated in a proof-of-concept experiment.

## 2   Related Work

In the supervised learning setting, a line of works have been done on multitask learning and representation learning with various assumptions [4, 15, 2, 5, 27, 8, 28, 14, 38]. These results assumed that all tasks share a joint representation function. It is also worth mentioning that [38] gave the method-of-moments estimator and built the confidence ball for the feature extractor, which inspired our algorithm for the infinite-action setting.

The benefit of representation learning has been studied in sequential decision-making problems, especially in RL domains. Arora et al. [3] proved that representation learning could reduce the sample complexity of imitation learning. D'eramo et al. [11] showed that representation learning could improve the convergence rate of the value iteration algorithm. Both require a probabilistic assumption similar to that in [28], and the statistical rates are of similar forms as those in [28]. Following these works, we study a special class of MDP called Linear MDP. Linear MDP [42, 21] is a popular model in RL, which uses linear function approximation to generalize large state-action space. [44] extends the definition to low inherent Bellman error (or IBE in short) MDPs. This model assumes that both the transition and the reward are near-linear in given features.

Recently, Yang et al. [40] showed multitask representation learning reduces the regret in linear bandits, using the framework developed by Du et al. [14]. Moreover, some works [17, 26, 21] proved results on the benefit of multitask representation learning RL with generative model or linear representation function. However, these works either restrict the representation function class to be linear, or the representation function is known to agent. This is unrealistic in real world practice, which limits these works' meaning.

The most relevant works that need to be mentioned is general function class value approximation for bandits and MDPs. Russo et al. [32] first proposed the concept of eluder dimension to measure the complexity of a function class and gave a regret bound for general function bandits using this dimension. Wang et al. [39] further proved that it can also be adopted in MDP problems. Dong et al. [12] extended the analysis with sequential Rademacher complexity. Inspired by these works, we adopt eluder dimension and develop our own analysis. But it should be pointed out that all those works focus on single task setting, which give a provable bound for just one single MDP or bandit problem. They lack the insight for why simultaneously dealing with multiple distinct but correlated tasks is more sample efficient. Our work aim to establish a framework to explain this. By considering locating the ground truth value function in multihead function space $\mathcal{F}^{\otimes M}$ (see detailed definition in section 4), we are able to theoretically explain the main reason for the boost of sample efficiency. Informally speaking, the shared feature extraction backbone $\phi$ receives samples from all the tasks, therefore accelerating the convergence for every single task compare with solving them separately.

## 3   Preliminaries

### 3.1   Notations

We use $[n]$ to denote the set $\{1, 2, \ldots, n\}$ and $\langle \cdot, \cdot \rangle$ to denote the inner product between two vectors. We use $f(x) = O(g(x))$ to represent $f(x) \leq C \cdot g(x)$ holds for any $x > x_0$ with some $C > 0$ and $x_0 > 0$. Ignoring the logarithm term, we use $f(x) = \tilde{O}(g(x))$ .

## 3.2 Multitask Contextual Bandits

We first study multitask representation learning in contextual bandits. Each task $i \in [M]$ is associated with an unknown function $f^{(i)} \in \mathcal{F}$ from certain function class $\mathcal{F}$. At each step $t \in [T]$, the agent is given a context vector $C_{t,i}$ from certain context space $\mathcal{C}$ and a set of actions $\mathcal{A}_{t,i}$ selected from certain action space $\mathcal{A}$ for each task $i$. The agent needs to choose one action $A_{t,i} \in \mathcal{A}_{t,i}$, and then receives a reward as $R_{t,i} = f^{(i)}(C_{t,i}, A_{t,i}) + \eta_{t,i}$, where $\eta_{t,i}$ is the random noise sampled from some i.i.d. distribution. The agent's goal is to understand function $f^{(i)}$ and maximize the cumulative reward, or equivalently, minimize the total regret from all $M$ tasks in $T$ steps defined as below.

$$\text{Reg}(T) \stackrel{\text{def}}{=} \sum_{t=1}^{T} \sum_{i=1}^{M} \left( f^{(i)}(C_{t,i}, A_{t,i}^{\star}) - f^{(i)}(C_{t,i}, A_{t,i}) \right),$$

where $A_{t,i}^{\star} = \arg\max_{A \in \mathcal{A}_{t,i}} f^{(i)}(C_{t,i}, A)$ is the optimal action with respect to context $C_{t,i}$ in task $i$.

## 3.3 Multitask MDP

Going beyond contextual bandits, we also study how this shared low-dimensional representation could benefit the sequential decision making problem like Markov Decision Process (MDP). In this work, we study undiscounted episodic finite horizon MDP problem. Consider an MDP $\mathcal{M} = (\mathcal{S}, \mathcal{A}, \mathcal{P}, r, H)$, where $\mathcal{S}$ is the state space, $\mathcal{A}$ is the action space, $\mathcal{P}$ is the transition dynamics, $r(\cdot, \cdot)$ is the reward function and $H$ is the planning horizon. The agent starts from an initial state $s_1$ which can be either fixed or sampled from a certain distribution, then interacts with environment for $H$ rounds. In the single task framework, at each round (also called level) $h$, the agent needs to perform an action $a_h$ according to a policy function $a_h = \pi_h(s_h)$. Then the agent will receive a reward $R_h(s_h, a_h) = r(s_h, a_h) + \eta_h$ where $\eta_h$ again is the noise term. The environment then transits the state from $s_h$ to $s_{h+1}$ according to distribution $\mathcal{P}(\cdot|s_h, a_h)$. The estimation for action value function given following action policy $\pi$ is defined as $Q_h^{\pi}(s_h, a_h) = r(s_h, a_h) + \mathbb{E}\left[\sum_{t=h+1}^{H} R_t(s_t, \pi_t(s_t))\right]$, and state value function is defined as $V_h^{\pi}(s_h) = Q_h^{\pi}(s_h, \pi_h(s_h))$. Note that there always exists a deterministic optimal policy $\pi^{\star}$ for which $V_h^{\pi^{\star}}(s) = \max_{\pi} V_h^{\pi}(s)$ and $Q_h^{\pi^{\star}}(s, a) = \max_{\pi} Q_h^{\pi}(s, a)$, we will denote them as $V_h^{\star}(s)$ and $Q_h^{\star}(s, a)$ for simplicity.

In the multitask setting, the agent gets a batch of states $\{s_{h,t}^{(i)}\}_{i=1}^{M}$ simultaneously from $M$ different MDP tasks $\{\mathcal{M}^{(i)}\}_{i=1}^{M}$ at each round $h$ in episode $t$, then performs a batch of actions $\{\pi_t^i(s_{h,t}^{(i)})\}_{i=1}^{M}$ for each task $i \in [M]$. Every $H$ rounds form an episode, and the agent will interact with the environment for totally $T$ episodes. The goal for the agent is minimizing the regret defined as

$$\text{Reg}(T) = \sum_{t=1}^{T} \sum_{i=1}^{M} V_1^{(i)\star}\left(s_{1,t}^{(i)}\right) - V_1^{\pi_t^i}\left(s_{1,t}^{(i)}\right),$$

where $V_1^{(i)\star}$ is the optimal value of task $i$ and $s_{1,t}^{(i)}$ is the initial state for task $i$ at episode $t$.

To let representation function play a role, it is assumed that all tasks share the same state space $\mathcal{S}$ and action space $\mathcal{A}$. Moreover, there exists a representation function $\phi : \mathcal{S} \times \mathcal{A} \mapsto \mathbb{R}^k$ such that action and state value function of all tasks $\mathcal{M}^{(i)}$ is always (approximately) linear in this representation. For example, given a representation function $\phi$, the action value approximation function at level $h$ is parametrized by a vector $\boldsymbol{\theta}_h \in \mathbb{R}^k$ as $Q_h[\phi, \boldsymbol{\theta}_h] \stackrel{\text{def}}{=} \langle \phi(s, a), \boldsymbol{\theta}_h \rangle$, similar for $V_h[\phi, \boldsymbol{\theta}_h](s) \stackrel{\text{def}}{=} \max_a \langle \phi(s, a), \boldsymbol{\theta}_h \rangle$. We denote all such action value functions as $\mathcal{Q}_h = \{Q_h[\phi, \boldsymbol{\theta}_h] : \phi \in \Phi, \boldsymbol{\theta}_h \in \mathbb{R}^k\}$, also value function approximation space as $\mathcal{V}_h = \{V_h[\phi, \boldsymbol{\theta}_h] : \phi \in \Phi, \boldsymbol{\theta}_h \in \mathbb{R}^k\}$. Each task $\mathcal{M}^{(i)}$ is a linear MDP, which means $\mathcal{Q}_h$ is always approximately close under Bellman operator $\mathcal{T}_h(Q_{h+1})(s, a) \stackrel{\text{def}}{=} r_h(s, a) + \mathbb{E}_{s' \sim \mathcal{P}_h(\cdot|s,a)} \max_{a'} Q_{h+1}(s', a')$.

**Linear MDP Definition.** *A finite horizon MDP $\mathcal{M} = (\mathcal{S}, \mathcal{A}, \mathcal{P}, r, H)$ is a linear MDP, if there exists a representation function $\phi : \mathcal{S} \times \mathcal{A} \mapsto \mathbb{R}^k$ and its induced value approximation function class $\mathcal{Q}_h, h \in [H]$, such that the inherent Bellman error*[44]

$$\mathcal{I}_h \stackrel{\text{def}}{=} \sup_{Q_{h+1} \in \mathcal{Q}_{h+1}} \inf_{Q_h \in \mathcal{Q}_h} \sup_{s \in \mathcal{S}, a \in \mathcal{A}} \left| (Q_h - \mathcal{T}_h(Q_{h+1}))(s, a) \right|,$$

*is always smaller than some small constant $\mathcal{I}$.*

The definition essentially assumes that for any Q-value approximation function $Q_{h+1} \in \mathcal{Q}_{h+1}$ at level $h + 1$, the Q-value function $Q_h$ at level $h$ induced by it can always be closely approximated in class $\mathcal{Q}_h$, which assures the accuracy through sequential levels.

### 3.4 Eluder Dimension

To measure the complexity of a general function class $f$, we adopt the concept of eluder dimension [32]. First, define $\epsilon$-dependence and independence.

**Definition 1 ($\epsilon$-dependent).** *An input $x$ is $\epsilon$-dependent on set $X = \{x_1, x_2, \ldots, x_n\}$ with respect to function class $\mathcal{F}$, if any pair of functions $f, \tilde{f} \in \mathcal{F}$ satisfying $\sqrt{\sum_{i=1}^{n}(f(x_i) - \tilde{f}(x_i))^2} \leq \epsilon$ also satisfies $|f(x) - \tilde{f}(x)| \leq \epsilon$. Otherwise, we call action $x$ to be $\epsilon$-independent of data set $X$.*

Intuitively, $\epsilon$-dependence captures the exhaustion of interpolation flexibility for function class $\mathcal{F}$. Given an unknown function $f$'s value on set $X = \{x_1, x_2, \ldots, x_n\}$, we are able to pin down its value on some particular input $x$ with only $\epsilon$-scale prediction error.

**Definition 2 ($\epsilon$-eluder dimension).** *The $\epsilon$-eluder dimension $\dim_E(\mathcal{F}, \epsilon)$ is the maximum length for a sequence of inputs $x_1, x_2, \ldots x_d \in \mathcal{X}$, such that for some $\epsilon' \geq \epsilon$, every element is $\epsilon'$-independent of its predecessors.*

This definition is similar to the definition of the dimensionality of a linear space, which is the maximum length of a sequence of vectors such that each one is linear independent to its predecessors. For instance, if $\mathcal{F} = \{f(x) : \mathbb{R}^d \mapsto \mathbb{R}, f(x) = \theta^\top x\}$, we have $\dim_E(\mathcal{F}, \epsilon) = O(d \log 1/\epsilon)$ since any $d$ linear independent input's estimated value can fully describe a linear mapping function. We also omit the $\epsilon$ and use $\dim_E(\mathcal{F})$ when it only has a logarithm dependent term on $\epsilon$.

## 4 Main Results for Contextual Bandits

In this section, we will present our theoretical analysis on the proposed GFUCB algorithm for contextual bandits.

### 4.1 Assumptions

This section will list the assumptions that we make for our analysis. The main assumption is the existence of a shared feature extraction function from class $\Phi = \{\phi : \mathcal{C} \times \mathcal{A} \mapsto \mathbb{R}^k\}$ that any task's value function is linear in this $\phi$.

**Assumption 1.1 (Shared Space and Representation)** *All the tasks share the same context space $\mathcal{C}$ and action space $\mathcal{A}$. Also, there exists a shared representation function $\phi \in \Phi$ and a set of $k$-dimensional parameters $\{\boldsymbol{\theta}_i\}_{i=1}^M$ such that each $f^{(i)}$ has the form $f^{(i)}(\cdot, \cdot) = \langle \phi(\cdot, \cdot), \boldsymbol{\theta}_i \rangle$.*

Following standard regularization assumptions for bandits [17, 40], we make assumptions on noise distribution and function parameters.

**Assumption 1.2 (Conditional Sub-Gaussian Noise)** *Denote $\mathcal{H}_{t,i} = \sigma(C_{1,i}, A_{1,i}, \ldots, C_{t,i}, A_{t,i})$ to be the $\sigma$-field summarizing the history information available before reward $R_{t,i}$ is observed for every task $i \in [M]$. We have $\eta_{t,i}$ is sampled from a 1-Sub-Gaussian distribution, namely $\mathbb{E}\left[\exp(\lambda \eta_{t,i}) \mid \mathcal{H}_{t,i}\right] \leq \exp\left(\frac{\lambda^2}{2}\right)$ for $\forall \lambda \in \mathbb{R}$*

**Assumption 1.3 (Bounded-Norm Feature and Parameter)** *We assume that the parameter $\boldsymbol{\theta}_i$ and the feature vector for any context-action pair $(C, A) \in \mathcal{C} \times \mathcal{A}$ is constant bounded for each task $i \in [M]$, namely $\|\boldsymbol{\theta}_i\|_2 \leq \sqrt{k}$ for $\forall i \in [M]$ and $\|\phi(C, A)\|_2 \leq 1$ for $\forall C \in \mathcal{C}, A \in \mathcal{A}$.*

Apart from these assumptions, we add assumption to measure and constrain the complexity of value approximation function class $\mathcal{F} = \mathcal{L} \circ \Phi$.

**Assumption 1.4 (Bounded Eluder Dimension).** *We assume that function class $\mathcal{F}$ has bounded Eluder dimension $d$, which means for any $\epsilon$, $\dim_E(\mathcal{F}, \epsilon) = \tilde{O}(d)$.*

## 4.2 Algorithm Details

---

**Algorithm 1** Generalized Functional UCB Algorithm

---
1: **for** step $t : 1 \to T$ **do**
2:     Compute $\mathcal{F}_t$ according to $(*)$
3:     Receive contexts $C_{t,i}$ and action sets $\mathcal{A}_{t,i}$, $i \in [M]$
4:     $f_t, A_{t,i} = \text{argmax}_{f \in \mathcal{F}_t, \, A_i \in \mathcal{A}_{t,i}} \sum_{i=1}^{M} f^{(i)}(C_{t,i}, A_i)$
5:     Play $A_{t,i}$ for task i, and get reward $R_{t,i}$ for $i \in [M]$.
6: **end for**

---

The details of the algorithm is in Algorithm 1. At each step $t$, the algorithm first solves the optimization problem below to get the empirically optimal solution $\hat{f}_t$ that best predicts the rewards for context-input pairs seen so far.

$$\hat{f}_t \leftarrow \underset{f \in \mathcal{F}^{\otimes M}}{\text{argmin}} \sum_{i=1}^{M} \sum_{k=1}^{t-1} \left( f^{(i)}(C_{k,i}, A_{k,i}) - R_{k,i} \right)^2$$

Here we abuse the notation of $\mathcal{F}^{\otimes M}$ as $\mathcal{F}^{\otimes M} = \left\{ f = \left( f^{(1)}, \ldots, f^{(M)} \right) : f^{(i)}(\cdot) = \phi(\cdot)^{\top} \boldsymbol{w}_i \in \mathcal{F} \right\}$ to denote the M-head prediction version of $\mathcal{F}$, parametrized by a shared representation function $\phi(\cdot)$ and a weight matrix $\boldsymbol{W} = [\boldsymbol{w}_1, \ldots, \boldsymbol{w}_M] \in \mathbb{R}^{k \times M}$. We use $f^{(i)}$ to denote the $i_{th}$ head of function $f$ which specially serves for task $i$.

After obtaining $\hat{f}_t$, we maintain a functional confidence set $\mathcal{F}_t \subseteq \mathcal{F}^{\otimes M}$ for possible value approximation functions

$$\mathcal{F}_t \overset{\text{def}}{=} \left\{ f \in \mathcal{F}^{\otimes M} : \left\| \hat{f}_t - f \right\|_{2, E_t}^2 \leq \beta_t, |f^{(i)}(\boldsymbol{x})| \leq 1, \forall \boldsymbol{x} \in \mathcal{C} \times \mathcal{A}, i \in [M] \right\} \qquad (*)$$

Here, for the sake of simplicity, we use $\left\| \hat{f}_t - f \right\|_{2, E_t}^2 = \sum_{i=1}^{M} \sum_{k=1}^{t-1} \left( \hat{f}_t^{(i)}(\boldsymbol{x}_{k,i}) - f^{(i)}(\boldsymbol{x}_{k,i}) \right)^2$ to denote the empirical 2-norm of function $\hat{f}_t - f = \left( \hat{f}_t^{(1)} - f^{(1)}, \ldots, \hat{f}_t^{(M)} - f^{(M)} \right)$. Basically, $(*)$ contains all the functions in $\mathcal{F}^{\otimes M}$ whose value estimation difference on all collected context-action pairs $\boldsymbol{x}_{k,i} = (C_{k,i}, A_{k,i})$ compared with empirical loss minimizer $\hat{f}_t$ does not exceed a preset parameter $\beta_t$. We show that with high probability, the real value function $f_\theta$ is always contained in $\mathcal{F}_t$ when $\beta_t$ is carefully chosen as $\tilde{O}(Mk + \log(\mathcal{N}(\Phi, \alpha, \| \cdot \|_\infty)))$, where $\mathcal{N}(\mathcal{F}, \alpha, \| \cdot \|_\infty)$ is the $\alpha$-covering number of function class $\Phi$ in the sup-norm $\|\phi\|_\infty = \max_{\boldsymbol{x} \in \mathcal{S} \times \mathcal{A}} \|\phi(\boldsymbol{x})\|_2$ and $\alpha$ is set to be a small number as $\frac{1}{kMT}$ (see detailed definition and proof in Lemma 1).

For the action choice, our algorithm follows *OFUL*, which estimates each action value with the most optimistic function value in our confidence set $\mathcal{F}_t$, and chooses the action whose optimistic value estimation is the highest. In the multitask setting, we choose one action from each task to form an action tuple $(A_1, A_2, \ldots, A_M)$ such that the summation of the optimistic value estimation $\sum_{i=1}^{M} f^{(i)}(C_{t,i}, A_i)$ is maximized by some function $f \in \mathcal{F}_t$.

**Intractability.** Some may have concerns on the intractability of building the confidence set $(*)$ and solving the optimization problem to get $\hat{f}_t, f_t, A_{t,i}$. The solution comes as two folds. From the theoretical perspective, since the focus of problem is sample complexity rather than computational complexity, a computational oracle can simply be assumed to give the solution of the optimization. This is the common practice for theoretical works [20, 35, 1, 19] in order to focus on the sample complexity analysis. From empirical perspective, there are great chances to optimize it with gradient methods. For example, solving $\hat{f}_t$ is a standard empirical risk minimization problem, and can be effectively solved with gradient methods [13]. As for $f_t$ and $A_{t,i}$, note that it is not necessary to explicitly build the confidence set $\mathcal{F}_t$ by listing all the candidates. The approximation algorithm just need to search within the confidence set via gradient method to optimize objective $\sum_{i=1}^{M} f^{(i)}(C_{t,i}, A_i)$. The start point is $\hat{f}_t$, and the algorithm knows that it approaches the border of $\mathcal{F}_t$ when $\|\hat{f}_t - f\|_{2, E_t}^2$ approaches $\beta_t$. The details of implementation are in section 6.

**Mechanism.** GFUCB algorithm solves the exploration problem in an implicit way. For a context-action pair $\boldsymbol{x} = (C, A)$ in task $i$ which has not been fully understood and explored yet, the possible value estimation $f^{(i)}(\boldsymbol{x})$ will vary in large range with regard to constraint $\|f - \hat{f}_t\|^2_{2, E_t} \leq \beta_t$. This is because within $\mathcal{F}_t$ there are many possible function value on this $\boldsymbol{x}$ while agreeing on all past context-action pairs' value. Therefore, the optimistic value $f^{(i)}(\boldsymbol{x})$ will become high by getting a significant implicit bonus, encouraging the agent to try such action $A$ under context $C$, which achieves natural exploration.

The reduction of sample complexity is achieved through joint training for function $\phi$. If we solve these tasks independently, the confidence set width $\beta_t$ is at scale $M \log\left(\mathcal{N}(\Phi, \alpha, \|\cdot\|_\infty)\right)$ because it needs to cover $M$ representation function space respectively. By involving $\phi$ in the prediction for all tasks, our algorithm reduces the size of confidence set by $M$ times, since now the samples from all the tasks can contribute to learn the representation $\phi$. Usually $\log\left(\mathcal{N}(\Phi, \alpha, \|\cdot\|_\infty)\right)$ is much greater than $k$ and $M$, hence our confidence set shrinks at a much faster speed. This explains how GFUCB achieves lower regret, since the sub-optimality at each step $t$ is proportional to the confidence set width $\beta_t$ when real value function $f_\theta \in \mathcal{F}_t$.

### 4.3 Regret Bound

Based on the assumptions above, we have the regret guarantee as below.

**Theorem 1.** *Based on assumption 1.1 to 1.4, denote the cumulative regret in $T$ steps as $\mathrm{Reg}(T)$, with probability at least $1 - \delta$ we have $\mathrm{Reg}(T) = \tilde{O}\left(\sqrt{MdT(Mk + \log \mathcal{N}(\Phi, \alpha_T, \|\cdot\|_\infty))}\right)$.*

Here, $d := \dim_E(\mathcal{F}, \alpha_T)$ is the Eluder dimension for value approximation function class $\mathcal{F} = \mathcal{L} \circ \Phi$, and $\alpha_T$ is discretization scale which only appears in logarithm term thus omitted. The detailed proof is left in appendix.

To the best of knowledge, this is the first regret bound for general function class representation learning in contextual bandits. To get a sense of its sharpness, note that when $\Phi$ is specialized as linear function class as $\Phi = \{\phi(x) = \boldsymbol{Bx}, \boldsymbol{B} \in \mathbb{R}^{k \times d}\}$, we have $\log \mathcal{N}(\Phi, \alpha_T, \|\cdot\|_\infty) = \tilde{O}(dk)$ and $\dim_E(\mathcal{F}) = d$, then our bound is reduced to $\tilde{O}(M\sqrt{dTk} + d\sqrt{MTk})$, which is the same optimal as the current best provable regret bound for linear representation class bandits in [17].

## 5 Main Results for MDP

### 5.1 Assumptions

For multitask Linear MDP setting, we adopt Assumption 3 from [17] which generalizes the inherent Bellman error [44] to multitask setting.

**Assumption 2.1 (Low IBE for multitask)** *Define multi-task IBE is defined as*

$$\mathcal{I}_h^{\mathrm{mul}} \stackrel{\mathrm{def}}{=} \sup_{\left\{Q_{h+1}^{(i)}\right\}_{i=1}^M \in \mathcal{Q}_{h+1}} \inf_{\left\{Q_h^{(i)}\right\}_{i=1}^M \in \mathcal{Q}_h} \sup_{s \in \mathcal{S}, a \in \mathcal{A}, i \in [M]} \left| \left(Q_h^{(i)} - \mathcal{T}_h^{(i)}\left(Q_{h+1}^{(i)}\right)\right)(s, a) \right|.$$

*We have $\mathcal{I} \stackrel{\mathrm{def}}{=} \sup_h \mathcal{I}_h^{\mathrm{mul}}$ is small for all $\mathcal{Q}_h$, $h \in [H]$.*

Assumption 2.1 generalize low IBE to multitask setting. It assumes that for every task $i \in [M]$, its Q-value function space is always close under Bellman operator.

**Assumption 2.2 (Parameter Regularization)** *We assume that*

- $\|\phi(s, a)\| \leq 1$, $0 \leq Q_h^\pi(s, a) \leq 1$ *for* $\forall (s, a) \in \mathcal{S} \times \mathcal{A}, h \in [H], \forall \pi$.

- *There exists a constant $D$ such that for any $h \in [H]$ and $\boldsymbol{\theta}_h^{(i)}$, it holds that $\|\boldsymbol{\theta}_h^{(i)}\|_2 \leq D$.*

- *For any fixed $\left\{Q_{h+1}^{(i)}\right\}_{i=1}^M \in \mathcal{Q}_{h+1}$, the random noise $z_h^{(i)} \stackrel{\mathrm{def}}{=} R_h^{(i)}(s, a) + \max_a Q_{h+1}^{(i)}(s', a) - \mathcal{T}_h^{(i)}\left(Q_{h+1}^{(i)}\right)(s, a)$ is bounded in $[-1, 1]$ and is always independent to all other random variables for $\forall (s, a) \in \mathcal{S} \times \mathcal{A}, h \in [H], i \in [M]$.*

These assumptions are widely adopted in linear MDP analytical works [44, 17, 26], which regularizes the parameter, feature, and noise scale. Again we add bounded Eluder dimension constraint for the Q-value estimation class.

**Assumption 2.3 (Bounded Eluder Dimension).** *We assume that function class $\mathcal{Q}_h$ has bounded Eluder dimension $d$ for any $h \in [H]$.*

## 5.2 Algorithm Details

---

**Algorithm 2** multitask Linear MDP Algorithm

---

1: **for** episode $t : 1 \to T$ **do**
2:     $Q_{H+1}^{(i)} = 0, i \in [M]$
3:     **for** $h : H \to 1$ **do**
4:        $\hat{\phi}_{h,t}, \hat{\boldsymbol{\theta}}_{h,t}^{(i)} \leftarrow$ solving (1)
5:        $Q_h^{(i)}(\cdot, \cdot) = \hat{\phi}_{h,t}(\cdot, \cdot)^\top \hat{\boldsymbol{\theta}}_{h,t}^{(i)}, V_h^{(i)}(\cdot) = \max_a Q_h^{(i)}(\cdot, a)$
6:     **end for**
7:     **for** $h : 1 \to H$ **do**
8:        Compute $\mathcal{F}_{h,t}$ according to Lemma 4
9:        Receive states $\left\{ s_{h,t}^{(i)} \right\}_{i=1}^M, \tilde{f}_{h,t}, a_{h,t}^{(i)} = \operatorname{argmax}_{f \in \mathcal{F}_{h,t}, a^{(i)} \in \mathcal{A}} \sum_{i=1}^M f^{(i)} \left( s_{h,t}^{(i)}, a^{(i)} \right)$
10:        Play $a_{h,t}^{(i)}$ and get reward $R_{h,t}^{(i)}$ for task $i \in [M]$.
11:     **end for**
12: **end for**

---

The algorithm for multitask linear MDP is similar to contextual bandits as above. The optimization problem in line 4 of Algorithm 2 is finding the empirically best solution for Q-value estimation at level $h$ in episode $t$ as below

$$
\hat{\phi}_{h,t}, \hat{\boldsymbol{\Theta}}_{h,t} \leftarrow \underset{\phi \in \Phi, \boldsymbol{\Theta} = [\boldsymbol{\theta}^{(1)}, \dots, \boldsymbol{\theta}^{(M)}]}{\operatorname{argmin}} \mathcal{L}(\phi, \boldsymbol{\Theta}) \tag{1}
$$
$$
s.t. \quad \|\boldsymbol{\theta}^{(i)}\| \le D, \forall i \in [M]
$$
$$
0 \le \phi(s, a)^\top \boldsymbol{\theta}_i \le 1, \forall (s, a) \in \mathcal{S} \times \mathcal{A}, i \in [M],
$$

where $\mathcal{L}(\phi, \boldsymbol{\Theta})$ is the empirical loss function defined as

$$
\sum_{i=1}^M \sum_{j=1}^{t-1} \left( \phi \left( s_{h,j}^{(i)}, a_{h,j}^{(i)} \right)^\top \boldsymbol{\theta}^{(i)} - R_{h,j}^{(i)} - V_{h+1}^{(i)} \left( s_{h+1,j}^{(i)} \right) \right)^2.
$$

The framework of our work resembles LSVI [21] and [26] which learns the Q-value estimation in a reverse order, at each level $h$, the algorithm uses just-learned value estimation function $V_{h+1}$ to build the regression target value as $R_{h,j}^{(i)} + V_{h+1}^{(i)} \left( s_{h+1,j}^{(i)} \right)$ and find empirically best estimation $\hat{f}_{h,t}^{(i)} = \hat{\phi}_{h,t}^\top \hat{\boldsymbol{\theta}}_{h,t}^{(i)}$ for each task $i \in [M]$. The optimistic value estimation of each action is again searched within confidence set $\mathcal{F}_{h,t}$ which centered at $\hat{f}_{h,t}$ and shrinks as the constraint $\|f - \hat{f}_{h,t}\|_{2,E_t}^2 \le \beta_t$ becomes increasingly tighter. Note that the contextual bandit problem can be regarded as a 1-horizon MDP problem without transition dynamics, and our framework at each level $h$ is indeed a copy of procedures in Algorithm 1.

## 5.3 Regret Bound

Based on assumptions 2.1 to 2.3, we prove that our algorithm enjoys a regret bound guaranteed by the following theorem. Detailed proof is left in appendix.

**Theorem 2.** *Based on assumption 2.1 to 2.3, denote the cumulative regret in $T$ episodes as $\operatorname{Reg}(T)$, we have the following regret bound for $\operatorname{Reg}(T)$ holds with probability at least $1 - \delta$ for Algorithm 2*

$$
\tilde{O} \left( MH\sqrt{Tdk} + H\sqrt{MTd \log \mathcal{N}(\Phi, \alpha)} + MHT\mathcal{I}\sqrt{d} \right),
$$

*where $\alpha$ is discretization scale smaller than $\frac{1}{kMT}$.*

**Remark.** Compared with naively executing single task general value function approximation algorithm [39] for $M$ tasks, whose regret bound is $\tilde{O}(MHd\sqrt{T\log\mathcal{N}(\Phi)})$, to achieve same average regret, our algorithm outperforms this naive algorithm with a boost of sample efficiency by $\tilde{O}(Md)$. This benefit mainly attributes to learning in function space $\mathcal{F}^{\otimes M} = \mathcal{L}^M \circ \Phi$ instead of $\mathcal{F}^M = (\mathcal{L} \circ \Phi)^M$, the former is more compact and requires much less samples to learn.

## 6 Experiments

To validate our theoretical findings, we conduct experiments on a non-linear neural network bandits. Note that it is a proof-of-concept experiment. Our main purpose is to realize the GFUCB algorithm and check its efficacy but *not* to beat sophisticated real-world algorithms. The point to demonstrate is that sample efficiency of GFUCB is scalable to the number of tasks and better than naive exploration.

### 6.1 Task Design

To test the efficacy of our algorithm, we use the MNIST dataset [10] to build a bandit problem that involves non-linear value approximation. The reward function of the bandit environment maps the same digit into the same base reward $r_b$, which ranges from 0 to 1, plus a noise $\eta_h$ sampled from a zero-mean Gaussian with a standard deviation of 0.01. At every round, each task will present the agent a context $C$ consists of $K$ different digit images and ask the agent to take action as an integer $j \in [K]$ meaning which image to choose, then return the reward according to the agent's choice.

For the multitask setting, we construct $M$ different tasks using different digit-to-reward mappings $\sigma_i : \{0, \ldots, 9\} \mapsto [0, 1], i \in [M]$, where $\sigma_i(k)$ will give a unique reward for all images of digit $k$ in task $i$. Different tasks have different reward mapping function $\sigma_i(\cdot)$. By designing the environment this way, it requires to learn a common representation $\phi$ to recognize digits for different tasks.

### 6.2 Implementation Details

We use a simple CNN as our feature extraction function $\phi$, which takes a digit image as input and outputs a 10-dimensional normalized vector as representation. It consists of two 3x3 convolution layers and two fully-connected layers, followed by ReLU activation and a normalization procedure.

The biggest challenge for implementation is how to solve a complex optimization problem in general functional space. In principle, finding parameters for a neural network to achieve the (near) minimal empirical error is an NP-Hard problem. To solve this issue, we use a gradient-based method to approximately find a local-optimal solution. For finding the empirically best $\hat{f}_t$, we use Adam with $lr = 1e-3$ to train for sufficiently long steps; in our setting, it is set to be 200 epochs at every step $t$, to ensure that the training loss is sufficiently low.

The next major challenge is estimating the optimistic value for each action within the abstract function set $\mathcal{F}_t$. To tackle this problem, we enumerate all possible action tuples $\{A_i\}_{i=1}^{M}$ and then solve the equivalent optimization below to compute its optimistic estimated value

$$\max_{f \in \mathcal{F}_t} \sum_{i=1}^{M} f^{(i)}(C_{t,i}, A_i) \quad s.t. \quad \left\| f - \hat{f}_t \right\|_{2,E_t}^2 \leq \beta_t.$$

Still, this is a complicated optimization problem within an abstract function set. Inspired by the Lagrangian operator, we transform it into an unconstrained optimization problem minimizing loss function $\ell(f) = -\sum_{i=1}^{M} f^{(i)}(A_i) + \lambda \cdot \max(0, \|\hat{f}_t - f\|_{2,E_t}^2 - B_t)$, where $\lambda$ is a hyperparameter to be determined, in our algorithm we set it to be $\lambda = 30$ by empirical search. Also $B_t = a\log(b \cdot t + c)$ is an approximation for $\beta_t$ since $\beta_t$ includes $\mathcal{N}(\Phi, \alpha)$ which is intractable to be exactly computed, we found $(a, b, c) = (0.4, 0.5, 2)$ to be a good parameter of UCB in single task. We use SGD with a small learning rate $(5e-4)$ to finetune the model $\hat{f}_t$ for 200 iterations to optimize $\ell(f)$.

The basic intuition is that, through optimizing $\ell(f)$, the algorithm will try to maximize function value $\sum_{i=1}^{M} f^{(i)}(A_i)$. And as long as $f$ satisfies $\|\hat{f}_t - f\|_{2,E_t}^2 \leq B_t$, such constraint will not appear in the loss term, thus has no effect on optimization. When $f$ comes to the border of $\mathcal{F}_t$, where

$\|\hat{f}_t - f\|^2_{2,E_t}$ approaches $B_t$, the second term adds regularization term to the loss as punishment, preserving $\|\hat{f}_t - f\|^2_{2,E_t}$ at a near-constant level around $B_t$. So we can approximately simulate the optimistic value estimating procedure via searching in the neighborhood of $\hat{f}_t$.

### 6.3  Connection to Algorithm 1

The main difference between our practical version algorithm and the theoretical one is that we did not list out all the functions in the whole confidence set $\mathcal{F}_t$ explicitly, but just use gradient-based method to implicitly search within a very small fraction of $\mathcal{F}_t$ with heuristics. Getting a candidate within the confidence set is much easier and tractable than rigorously exhausting all functions in $\mathcal{F}_t$ to optimize. We can start from the parameter of $\hat{f}_t$ and use gradient method to approximately find $f_t$ and $A_{t,i}$.

Another difference is we do not rigorous compute $\beta_t$ which involves $\mathcal{N}(\Phi)$, but directly determine a parametrized function form. Rigorously speaking, our tuned value of $\beta_t$ is much smaller than the theoretical guaranteed ones, so all the candidate functions that we search along the trajectory of gradient method still satisfy the theoretical requirement (but it may omit many other potential candidates). Therefore, our practical version algorithm should be regarded as an inaccurate approximation to the theoretical algorithm. Moreover, it also plays a role as regularization to enable the convergence of $\mathcal{F}_t$ since we only consider regular ones in the neighborhood of $\hat{f}_t$.

### 6.4  Results

We test the performance of our algorithm against a naive eps-greedy baseline that solves each task independently by training *the same* CNN value prediction module. We show our results with number of tasks $M = 1, 5, 10$ in Figure 1. Firstly, we randomly generate 10 different digit-value mapping functions $\sigma_i(\cdot), i = 1, \ldots, 10$. The total 10 tasks are divided into $10/M$ groups; each group forms a $M$-task problem and is solved by an individual copy of some algorithm. At each step $t$, the cumulative regret from all 10 tasks is averaged to estimate the method's performance. Our result in Figure 1 verified that the multitask training does accelerate learning, which empirically validates our theoretical analysis. The multitask training utilizes the samples from all $M$ tasks to jointly learn a good representation $\phi$, which significantly accelerates the learning procedure of the CNN backbone. Also, the improvement in GFUCB algorithm's performance with $M = 1$ validates the effect of our finetune procedure for getting a bonus. Detailed dissection and discussion are left in appendix.

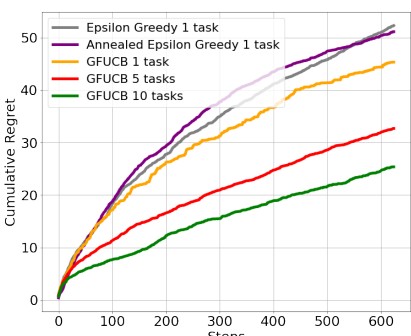

Figure 1: Cumulative regret over steps for $M = 1, 5, 10$.

## 7  Conclusion

In this work, we extend the analysis of the benefit of multitask representation learning from linear representation class to general function class. We propose a straightforward algorithm that can utilize samples from all the tasks to jointly train a representation function, which is demonstrated theoretically and empirically to accelerate the sample efficiency and outperform naively single-task learning. Also, we extend the analysis to the MDP setting and show that the benefit of multitask representation learning is similar. Furthermore, our experimental result reveals that our proposed algorithm is also effective in practice even for highly non-linear neural network representations.

## Acknowledgments and Disclosure of Funding

This work is supported in part by the National Science and Technology Major Project of the Ministry of Science and Technology of China under Grants 2018AAA0101604, the National Natural Science Foundation of China under Grants 62022048 and the State Key Lab of Autonomous Intelligent Unmanned Systems.

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
