## A  Bandit Regret Bound Analysis

### A.1  Algorithm Procedure

At each round $s \in [t]$, after performing a list of actions $\{A_{s,i}\}_{i=1}^{M}$ with respect to corresponding context vectors $\{C_{s,i}\}_{i=1}^{M}$, the agent receives a list of rewards $y_{s,i}$ associated with input $\boldsymbol{x}_{s,i} = (C_{s,i}, A_{s,i})$ for $i \in [M]$. Note that we will use $f(C_t, A_t)$ or $f(\boldsymbol{x}_t)$ where $\boldsymbol{x}_t = (C_t, A_t)$ in different contexts. The algorithm first solves the following regression problem to obtain the empirical minimizer function $\hat{f}_t(\cdot) = \hat{\phi}_t(\cdot)^{\top} \widehat{\boldsymbol{W}}_t$ based on samples collected.

$$\hat{\phi}_t, \widehat{\boldsymbol{W}}_t = \operatorname*{argmin}_{\phi \in \Phi, \boldsymbol{W} = [\boldsymbol{w}_1, \dots, M]} \sum_{i=1}^{M} \left\| \boldsymbol{y}_{t-1,i} - \phi(\boldsymbol{X}_{t-1,i})^{\top} \boldsymbol{w}_i \right\|_2^2$$

$$s.t. \quad |\phi(\boldsymbol{x})^{\top} \boldsymbol{w}_i| \leq 1, \quad \forall i \in [M], \boldsymbol{x} \in \mathcal{C} \times \mathcal{A}.$$

Here, $\boldsymbol{X}_{t-1,i} = [\boldsymbol{x}_{1,i}, \boldsymbol{x}_{2,i}, \dots, \boldsymbol{x}_{t-1,i}]$ is the selected context-action pair for task $i$ in the first $t-1$ rounds, and $\boldsymbol{y}_{t-1,i} = [R_{1,i}, R_{2,i}, \dots, R_{t-1,i}]^{\top} \in \mathbb{R}^{t-1}$ stacks all the received reward into a vector accordingly. We use $\phi(\boldsymbol{X})$ to compactly represent feeding each column $\boldsymbol{x}_i$ of $\boldsymbol{X}$ into $\phi(\cdot)$ and get concatenated output as $[\phi(\boldsymbol{x}_1), \phi(\boldsymbol{x}_2), \dots, \phi(\boldsymbol{x}_{t-1})]$.

After obtaining the best empirical estimator function $\hat{f}_t^{(i)}(\cdot) = \hat{\phi}_t(\cdot)^{\top} \hat{\boldsymbol{w}}_{t,i}$ at round $t \in [T]$ for each $i \in [M]$, we maintain a function confidence set $\mathcal{F}_t \subseteq \mathcal{F}^{\otimes M}$ for representation function and parameters.

$$\mathcal{F}_t \stackrel{\text{def}}{=} \left\{ f \in \mathcal{F}^{\otimes M} : \left\| \hat{f}_t - f \right\|_{2, E_t}^2 \leq \beta_t, |f^{(i)}(\boldsymbol{x})| \leq 1, \forall \boldsymbol{x} \in \mathcal{C} \times \mathcal{A}, i \in [M] \right\} \qquad (*)$$

Here we abuse the notation of $\mathcal{F}^{\otimes M}$ as $\mathcal{F}^{\otimes M} = \left\{ f = \left( f^{(1)}, \dots, f^{(M)} \right) : f^i(\cdot) = \phi(\cdot)^{\top} \boldsymbol{w}_i \in \mathcal{F} \right\}$ to denote the M-head prediction version of $\mathcal{F}$, parametrized by a shared representation function $\phi(\cdot)$ and a weight matrix $\boldsymbol{W} = [\boldsymbol{w}_1, \dots, \boldsymbol{w}_M] \in \mathbb{R}^{k \times M}$. We use $f^{(i)}$ to denote the $i_{th}$ head of function $f$. For the sake of simplicity, we use

$$\left\| \hat{f}_t - f \right\|_{2, E_t}^2 = \sum_{i=1}^{M} \sum_{s=1}^{t-1} \left( \hat{f}_t^{(i)}(\boldsymbol{x}_{s,i}) - f^{(i)}(\boldsymbol{x}_{s,i}) \right)^2$$

to denote the empirical 2-norm of function $\hat{f}_t - f = \left( \hat{f}_t^{(1)} - f^{(1)}, \dots, \hat{f}_t^{(M)} - f^{(M)} \right)$. Another important hyperparameter for our algorithm is the confidence set width term $\beta_t$, which is a function of representation function class $\Phi$, probability $\delta$ and discretization scale parameter $\alpha$.

$$\beta_t(\Phi, \alpha, \delta) = 12Mk + 12 \log \left( \mathcal{N}(\Phi, \alpha, \| \cdot \|_{\infty}) / \delta \right) + 8\alpha \sqrt{Mtk(Mt + \log(2Mt^2/\delta))}$$

here $\mathcal{N}(\mathcal{F}, \alpha, \| \cdot \|_{\infty})$ is the $\alpha$-covering number of function class $\Phi$ in the sup-norm $\|\phi\|_{\infty} = \max_{\boldsymbol{x} \in \mathcal{S} \times \mathcal{A}} \|\phi(\boldsymbol{x})\|_2$ (see detailed definition in Lemma 1) and $\alpha$ can be set to be some small scale number, like $\frac{1}{kMT}$.

### A.2  Main Proof sketch

In this section we will give a theoretical guarantee for the performance of our algorithm. Before diving into details, we first explain the overall idea and structure of our proof. First, we decompose the regret into the summation of confidence set width at different rounds plus a small term which accounts for the possibility that confidence function set $\mathcal{F}_t$ fails to contain ground truth function $f_\theta$.

**Lemma 0.** *Fix any sequence of confidence set $\{\mathcal{F}_t, t \in \mathbb{N}\}$ which is measurable with respect to history $\mathcal{H}_t$, denote the induced policy by Algorithm 1 as $\pi = \{\pi_i\}_{i=1}^{M}$ where each $\pi_i : \mathcal{C} \mapsto \mathcal{A}, i \in [M]$ is for task i, then for any $T \in \mathbb{N}$ we have*

$$\operatorname{Regret}(T) := \sum_{i=1}^{M} \sum_{t=1}^{T} f_\theta^{(i)}\left( \boldsymbol{x}_{t,i}^{\star} \right) - f_\theta^{(i)}(\boldsymbol{x}_{t,i}) \leq \sum_{t=1}^{T} \left[ w_{\mathcal{F}_t}(\boldsymbol{X}_t) + C \cdot \mathbb{I}(f_\theta \notin \mathcal{F}_t) \right]$$

where $\boldsymbol{x}_{t,i} = (C_{t,i}, \pi_i(C_{t,i}))$ is the context-action pair that actually happened. $A_{t,i}^\star = \arg\max_A f_\theta^{(i)}(C_{t,i}, A)$ is the optimal action for each task $i \in [M]$ at round $t \in [T]$, and $\boldsymbol{x}_{t,i}^\star = (C_{t,i}, A_{t,i}^\star)$ is the corresponding optimal context-action pair, $C$ is a universal large enough constant. We use $\boldsymbol{X}_t = [\boldsymbol{x}_{t,1}, \ldots, \boldsymbol{x}_{t,M}]$ to stack $\boldsymbol{x}_{t,i}$ into a matrix, similar for $\boldsymbol{X}_t^\star = [\boldsymbol{x}_{t,1}^\star, \ldots, \boldsymbol{x}_{t,M}^\star]$. The confidence set width $w_{\mathcal{F}_t}(\boldsymbol{X}_t)$ is defined by

$$w_{\mathcal{F}_t}(\boldsymbol{X}_t) := \sup_{\overline{f}, \underline{f} \in \mathcal{F}_t} \sum_{i=1}^M \left[ \overline{f}^{(i)}(\boldsymbol{x}_{t,i}) - \underline{f}^{(i)}(\boldsymbol{x}_{t,i}) \right].$$

Essentially, it measures the largest total difference of value estimation among all the functions in $f \in \mathcal{F}_t$ for the fixed inputs $\boldsymbol{x}_{t,i}$ where $i \in [M]$. Apart from the constant term accounting for the case that $\mathcal{F}_t$ fails to contain $f_\theta$, which we will prove happen with small probability, this regret is then bounded by the sum of width over time step $t$.

Next, we will show that our construction of confidence set $\mathcal{F}_t$ makes all of them contain real value function with high probability.

**Lemma 1.** *For all $\delta \in (0,1)$ and $\alpha > 0$, if $\mathcal{F}_t$ is defined by $\mathcal{F}_t = \{f \in \mathcal{F}^{\otimes M} : \|f - \hat{f}\|_{2, E_t} \leq \sqrt{\beta_t(\Phi, \delta, \alpha)}\}$ for all $t \in \mathbb{N}$, where $\hat{f}$ is the solution to the empirical error minimization. Denote the ground truth value function as $f_\theta(\cdot)$, then we have*

$$\mathbb{P}\left( f_\theta \in \bigcap_{t=1}^T \mathcal{F}_t \right) \geq 1 - 2\delta.$$

After that, we prove that

**Lemma 2.**
$$\sum_{t=1}^T \mathbb{I}\left(w_{\mathcal{F}_t}(\boldsymbol{X}_t) > \epsilon\right) \leq \left( \frac{4M\beta_T}{\epsilon^2} + 1 \right) \dim_E(\mathcal{F}, \epsilon)$$

Then plug it into lemma 0, we get our main result for the regret bound as

$$\mathrm{Reg}(\pi, T) \leq \frac{1}{T} + \min\left\{\dim_E(\mathcal{F}, \alpha_T), T\right\} + 4\sqrt{M \dim_E(\mathcal{F}, \alpha_T)\beta_T T} \tag{1}$$

Usually $\alpha_T$ is set to be a small number like $\frac{1}{kMT}$, or the minimizer for $\beta_T(\Phi, \alpha, \delta)$. We know that $\dim_E(\mathcal{F}, \alpha_T)$ is a poly-logarithmic function of $T$, which means the final regret bound is dominant by term $\sqrt{M \dim_E(\mathcal{F}, \alpha_T)\beta_T T}$ when $T \to \infty$. This further becomes

$$\sqrt{MT\left(Mk + \log\left(\mathcal{N}(\Phi, (kMT)^{-1}, \|\cdot\|_\infty)\right)\right)\dim_E(\mathcal{F}, (kMT)^{-1})} \tag{2}$$

For example, if $\Phi$ is specialized as linear function class parametrized by matrix $\boldsymbol{\Theta} \in \mathbb{R}^{d \times k}$, then $\log\left(\mathcal{N}(\Phi, (kMT)^{-1}, \|\cdot\|_\infty)\right) = O(kd\log(kMT))$ and $\dim_E(\mathcal{F}, (kMT)^{-1}) = O(d\log(kMT))$, hence the regret bound becomes

$$O(\sqrt{MT(Mk + kd)d}\log(kMT)) = \tilde{O}(M\sqrt{kdT} + d\sqrt{MkT})$$

which reduces to result in [**?**] by a poly-logarithm factor.

## A.3 Detailed Proof

*Proof of Lemma 0.* Define the upper and lower bounds $U_t(\boldsymbol{X}_t) = \sup\left\{\sum_{i=1}^M f^{(i)}(\boldsymbol{x}_{t,i}) : f \in \mathcal{F}_t\right\}$ and $L_t(\boldsymbol{X}_t) = \inf\left\{\sum_{i=1}^M f^{(i)}(\boldsymbol{x}_{t,i}) : f \in \mathcal{F}_t\right\}$.

If $f_\theta \notin \mathcal{F}_t$, then the error will be bounded by a large constant $C$ since all $f(\boldsymbol{x})$ is constant bounded. Otherwise $f_\theta \in \mathcal{F}_t$, we have

$$L_t(\boldsymbol{X}_t) \leq \sum_{i=1}^M f_\theta^{(i)}(\boldsymbol{x}_{t,i}) \leq U_t(\boldsymbol{X}_t)$$

$$\sum_{i=1}^{M} f_\theta^{(i)}(\boldsymbol{x}_{t,i}^\star) \le U_t(\boldsymbol{X}_t^\star)$$

where $\boldsymbol{X}_t$ and $\boldsymbol{X}_t^\star$ is defined in lemma 0. Also, by the optimality of $\boldsymbol{X}_t$ with respect to $\mathcal{F}_t$, we know $U_t(\boldsymbol{X}_t^\star) \le U_t(\boldsymbol{X}_t)$, therefore

$$
\begin{aligned}
\sum_{i=1}^{M} \left[ f_\theta^{(i)}(\boldsymbol{x}_{t,i}^\star) - f_\theta^{(i)}(\boldsymbol{x}_{t,i}) \right] \le & C \cdot \mathbb{I}(f_\theta \notin \mathcal{F}_t) + [U_t(\boldsymbol{X}_t^\star) - L_t(\boldsymbol{X}_t)] \\
= & C \cdot \mathbb{I}(f_\theta \notin \mathcal{F}_t) + \sum_{i=1}^{M} [U_t(\boldsymbol{X}_t^\star) - U_t(\boldsymbol{X}_t) + U_t(\boldsymbol{X}_t) - L_t(\boldsymbol{X}_t)] \\
\le & C \cdot \mathbb{I}(f_\theta \notin \mathcal{F}_t) + \sum_{i=1}^{M} [U_t(\boldsymbol{X}_t) - L_t(\boldsymbol{X}_t)] \\
= & C \cdot \mathbb{I}(f_\theta \notin \mathcal{F}_t) + w_{\mathcal{F}_t}(\boldsymbol{X}_t)
\end{aligned}
$$

Take summation over $t \in [T]$ and complete the proof. $\qquad\square$

**Lemma 1.** *For all $\delta \in (0,1)$ and $\alpha > 0$, if $\mathcal{F}_t$ is defined by $\mathcal{F}_t = \left\{ f \in \mathcal{F}^{\otimes M} : \|f - \hat{f}\|_{2,E_t} \le \sqrt{\beta_t(\Phi, \delta, \alpha)} \right\}$ for all $t \in \mathbb{N}$, where $\hat{f}$ is the solution to the empirical error minimization. Denote the ground truth value function as $f_\theta$, then we have*

$$\mathbb{P}\left( f_\theta \in \bigcap_{t=1}^{T} \mathcal{F}_t \right) \ge 1 - 2\delta.$$

*Proof of Lemma 1.* Denote $L_{2,t}(f) = \sum_{i=1}^{M} \sum_{s=1}^{t} |f^{(i)}(\boldsymbol{x}_{s,i}) - y_{s,i}|^2$ and $\tilde{f}_t = \hat{f}_t - f_\theta$, we have

$$
\begin{aligned}
L_{2,t}(\hat{f}) - L_{2,t}(f_\theta) &= \sum_{i=1}^{M} \sum_{s=1}^{t} \left| \hat{f}_t^{(i)}(\boldsymbol{x}_{s,i}) - y_{s,i} \right|^2 - \left| f_\theta^{(i)}(\boldsymbol{x}_{s,i}) - y_{s,i} \right|^2 & (3) \\
&= \sum_{i=1}^{M} \sum_{s=1}^{t} \left| \hat{f}_t^{(i)}(\boldsymbol{x}_{s,i}) - f_\theta^{(i)}(\boldsymbol{x}_{s,i}) - \eta_{s,i} \right|^2 - \eta_{s,i}^2 & (4) \\
&= \left\| \hat{f}_t - f_\theta \right\|_{2,E_t}^2 - \sum_{i=1}^{M} \sum_{s=1}^{t} 2\eta_{s,i} \cdot \tilde{f}_t^{(i)}(\boldsymbol{x}_{s,i}) & (5)
\end{aligned}
$$

By the optimality of $\hat{f}$, we know $(5) \le 0$, hence

$$\left\| \hat{f}_t - f_\theta \right\|_{2,E_t}^2 \le \sum_{i=1}^{M} 2 \left\langle \boldsymbol{\eta}_{t,i}, \tilde{f}_t^{(i)}(\boldsymbol{X}_{t,i}) \right\rangle \qquad (6)$$

here $\tilde{f}_t^{(i)}(\boldsymbol{X}_{t,i}) = [\tilde{f}_t^{(i)}(\boldsymbol{x}_{1,i}), \tilde{f}_t^{(i)}(\boldsymbol{x}_{2,i}), \ldots, \tilde{f}_t^{(i)}(\boldsymbol{x}_{t,i})]^\top$ and $\boldsymbol{\eta}_{t,i} = [\eta_{1,i}, \eta_{2,i}, \ldots, \eta_{t,i}]^\top$ are both in $\mathbb{R}^t$. We can represent each function $\tilde{f}_t^{(i)}(\cdot)$ in form $\tilde{f}_t^{(i)}(\cdot) = \left[ \phi^\star(\cdot)^\top, \hat{\phi}_t(\cdot)^\top \right] \begin{bmatrix} \boldsymbol{w}_{t,i}^\star \\ -\hat{\boldsymbol{w}}_{t,i} \end{bmatrix} = \phi^\star(\cdot)^\top \boldsymbol{w}_{t,i}^\star - \hat{\phi}_t(\cdot)^\top \hat{\boldsymbol{w}}_{t,i}$, which is exactly $f_\theta - \hat{f}_t$. Denote $\tilde{\phi}_t(\cdot) = \begin{bmatrix} \phi^\star(\cdot) \\ \hat{\phi}_t(\cdot) \end{bmatrix} \in \Phi^2$ and $\tilde{\boldsymbol{w}}_{t,i} = \begin{bmatrix} \boldsymbol{w}_{t,i}^\star \\ -\hat{\boldsymbol{w}}_{t,i} \end{bmatrix} \in \mathbb{R}^{2k}$, then $\tilde{f}_t^{(i)}(\cdot) = \tilde{\phi}_t(\cdot)^\top \tilde{\boldsymbol{w}}_{t,i}$. Since the output of $\tilde{\phi}_t(\boldsymbol{x}_{s,i}) \in \mathbb{R}^{2k}$, we can take following decomposition for each $i \in [M]$

$$\tilde{\phi}_t(\boldsymbol{X}_{t,i}) = \left[ \tilde{\phi}_t(\boldsymbol{x}_{s,i}) \right]_{s=1}^{t}, \quad \tilde{\phi}_t(\boldsymbol{X}_{t,i})^\top = \boldsymbol{U}_i \boldsymbol{Q}_i, \quad \boldsymbol{U}_i \in \mathcal{O}^{t \times 2k}, \boldsymbol{Q}_i \in \mathbb{R}^{2k \times 2k}.$$

For regret bound, we only need to care about $t \geq 2k$ by a constant regret difference, hence this decomposition is possible. Plug it into (6) and we get

$$\frac{1}{2}\left\|\hat{f} - f_\theta\right\|_{2,E_t}^2 \leq \sum_{i=1}^{M}\left\langle \boldsymbol{\eta}_{t,i}, \tilde{f}_t^{(i)}(\boldsymbol{X}_{t,i})\right\rangle \tag{7}$$

$$= \sum_{i=1}^{M} \boldsymbol{\eta}_{t,i}^\top \cdot \tilde{\phi}_t(\boldsymbol{X}_{t,i})^\top \tilde{\boldsymbol{w}}_{t,i} \tag{8}$$

$$= \sum_{i=1}^{M} \boldsymbol{\eta}_{t,i}^\top \cdot \boldsymbol{U}_i \boldsymbol{Q}_i \tilde{\boldsymbol{w}}_{t,i} \tag{9}$$

Notice that, however, $\boldsymbol{U}_t$ is obtained from optimization problem, which further depends on concrete sampled noise $\boldsymbol{\eta}_{t,i}$, hence the concentration bound based on i.i.d. assumption cannot be applied directly. If we fix function $\tilde{f}_t = \bar{f}_t$, which induces corresponding $\bar{\phi}_t(\cdot)$ and $\bar{\phi}_t(\boldsymbol{X}_{t,i}) = \bar{\boldsymbol{U}}_i(\bar{\phi})\bar{\boldsymbol{Q}}_i$, $\bar{\boldsymbol{U}}_i(\bar{\phi})$ means $\bar{\boldsymbol{U}}_i$ is a function determined by $\bar{\phi}$. According to standard sub-exponential random variable concentration bound, each $\bar{\boldsymbol{U}}_i(\bar{\phi})$ has $2k$ independent degrees of freedom, hence we know that with probability at least $1 - \delta_1$

$$\sum_{i=1}^{M}\|\bar{\boldsymbol{U}}_i^\top \boldsymbol{\eta}_{t,i}\|^2 \leq 2Mk + \log(1/\delta_1) \tag{10}$$

Denote $\Phi^2 = \{g(\boldsymbol{x}) = [\phi_1(\boldsymbol{x})^\top, \phi_2(\boldsymbol{x})^\top]^\top : \phi_1, \phi_2 \in \Phi\}$, $\Phi_\alpha^2$ is an $\alpha$-cover of $\Phi^2$ such that for any $\phi \in \Phi^2$, there is a $\phi_\alpha \in \Phi_\alpha^2$ such that

$$\max_{\boldsymbol{x} \in \mathcal{C} \times \mathcal{A}} \|\phi(\boldsymbol{x}) - \phi_\alpha(\boldsymbol{x})\|_2 \leq \alpha. \tag{11}$$

For $\tilde{\phi}$, find a closest $\bar{\phi} \in \Phi_\alpha^2$ from $\alpha$-cover net to satisfy the requirement above, then denote $\bar{f}_t^{(i)}(\cdot) = \bar{\phi}(\cdot)^\top \tilde{\boldsymbol{w}}_{t,i}$. By union bound, we know that with probability at least $1 - |\Phi_\alpha^2|\delta_1$, for any $\bar{\phi} \in \Phi_\alpha^2$, the induced $\bar{\boldsymbol{U}}_i(\bar{\phi})$ satisfy inequality (10), therefore

$$\frac{1}{2}\left\|\hat{f}_t - f_\theta\right\|_{2,E_t}^2 \leq \sum_{i=1}^{M}\left\langle \boldsymbol{\eta}_{t,i}, \tilde{f}_t^{(i)}(\boldsymbol{X}_{t,i})\right\rangle \tag{12}$$

$$= \sum_{i=1}^{M} \boldsymbol{\eta}_{t,i}^\top \cdot \boldsymbol{U}_i \boldsymbol{Q}_i \tilde{\boldsymbol{w}}_{t,i} = \sum_{i=1}^{M} \boldsymbol{\eta}_{t,i}^\top \cdot (\boldsymbol{U}_i - \bar{\boldsymbol{U}}_i + \bar{\boldsymbol{U}}_i)\boldsymbol{Q}_i \tilde{\boldsymbol{w}}_{t,i} \tag{13}$$

$$= \sum_{i=1}^{M} \boldsymbol{\eta}_{t,i}^\top \cdot \bar{\boldsymbol{U}}_i \boldsymbol{Q}_i \tilde{\boldsymbol{w}}_{t,i} + \sum_{i=1}^{M} \boldsymbol{\eta}_{t,i}^\top \cdot (\boldsymbol{U}_i - \bar{\boldsymbol{U}}_i)\boldsymbol{Q}_i \tilde{\boldsymbol{w}}_{t,i} \tag{14}$$

$$\leq \sqrt{\sum_{i=1}^{M}\left\|\bar{\boldsymbol{U}}_i^\top \boldsymbol{\eta}_{t,i}\right\|^2} \cdot \sqrt{\sum_{i=1}^{M}\|\boldsymbol{Q}_i \tilde{\boldsymbol{w}}_{t,i}\|^2} + \sum_{i=1}^{M}\left\langle \boldsymbol{\eta}_{t,i}, \tilde{f}_t - \bar{f}_t\right\rangle \tag{15}$$

$$\leq \sqrt{\sum_{i=1}^{M}\left\|\bar{\boldsymbol{U}}_i^\top \boldsymbol{\eta}_{t,i}\right\|^2} \cdot \sqrt{\sum_{i=1}^{M}\|\boldsymbol{U}_i \boldsymbol{Q}_i \tilde{\boldsymbol{w}}_{t,i}\|^2} + \sum_{i=1}^{M}\left\langle \boldsymbol{\eta}_{t,i}, \tilde{f}_t - \bar{f}_t\right\rangle \tag{16}$$

$$= \sqrt{\sum_{i=1}^{M}\left\|\bar{\boldsymbol{U}}_i^\top \boldsymbol{\eta}_{t,i}\right\|^2} \cdot \left\|\tilde{f}\right\|_{2,E_t} + \sum_{i=1}^{M}\left\langle \boldsymbol{\eta}_{t,i}, \tilde{f}_t - \bar{f}_t\right\rangle \tag{17}$$

$$\leq \sqrt{2Mk + \log(1/\delta_1)} \cdot \left\|\tilde{f}\right\|_{2,E_t} + \sqrt{\sum_{i=1}^{M}\|\boldsymbol{\eta}_{t,i}\|^2} \cdot \left\|\tilde{f}_t - \bar{f}_t\right\|_{2,E_t} \tag{18}$$

The first term of (18) comes from (10), and the second term is from Cauchy inequality. We assign $\delta_t = \frac{\delta_2}{T}$ failure probability for event

$$\omega_t : \sum_{i=1}^{M}\|\boldsymbol{\eta}_{t,i}\|^2 \geq Mt + \log(2Mt/\delta_t).$$

By union bound, we have

$$\mathbb{P}\left(\exists t \in [T] : \sum_{i=1}^{M} \|\boldsymbol{\eta}_{t,i}\|^2 \ge Mt + \log(2Mt^2/\delta_2)\right) \le \sum_{t=1}^{T} \delta_t \le \delta_2. \tag{19}$$

Next we will give a bound for $\|\tilde{f}_t - \bar{f}_t\|_{2,E_t}$.

$$\left\|\tilde{f}_t - \bar{f}_t\right\|_{2,E_t}^2 = \sum_{i=1}^{M} \sum_{s=1}^{t} \left| \tilde{\phi}_t(\boldsymbol{x}_{s,i})^\top \tilde{\boldsymbol{w}}_{s,i} - \bar{\phi}_t(\boldsymbol{x}_{s,i})^\top \tilde{\boldsymbol{w}}_{s,i} \right|^2 \tag{20}$$

$$= \sum_{i=1}^{M} \sum_{s=1}^{t} \left| (\tilde{\phi}_t(\boldsymbol{x}_{s,i}) - \bar{\phi}_t(\boldsymbol{x}_{s,i}))^\top \tilde{\boldsymbol{w}}_{s,i} \right|^2 \tag{21}$$

$$\le \sum_{i=1}^{M} \sum_{s=1}^{t} \left\| \tilde{\phi}_t(\boldsymbol{x}_{s,i}) - \bar{\phi}_t(\boldsymbol{x}_{s,i}) \right\|_2^2 \cdot \|\tilde{\boldsymbol{w}}_{s,i}\|_2^2 \tag{22}$$

According to our assumption, we know $\|\tilde{\boldsymbol{w}}_{s,i}\|^2 \le 2\|\boldsymbol{w}_{s,i}\|^2 + 2\|\hat{\boldsymbol{w}}_{s,i}\|^2 \le 4k$, from (11) we know $\left\| \tilde{\phi}_t(\boldsymbol{x}_{s,i}) - \bar{\phi}_t(\boldsymbol{x}_{s,i}) \right\|_2 \le \alpha$, hence

$$\left\|\tilde{f}_t - \bar{f}_t\right\|_{2,E_t}^2 \le 4Mtk\alpha^2 \tag{23}$$

Plug (19) and (23) back into (18), we know with probability at least $1 - \delta_2 - |\Phi_\alpha^2|\delta_1$, for any $t \in \mathbb{N}$

$$\frac{1}{2}\left\|\tilde{f}_t\right\|_{2,E_t}^2 \le \sqrt{2Mk + \log(1/\delta_1)} \cdot \left\|\tilde{f}_t\right\|_{2,E_t} + \sqrt{Mt + \log(2Mt^2/\delta_2)} \cdot \sqrt{4Mtk\alpha^2} \tag{24}$$

Some simple algebraic transform gives

$$\left\|\hat{f}_t - f_\theta\right\|_{2,E_t}^2 = \left\|\tilde{f}_t\right\|_{2,E_t}^2 \le 6(2Mk + \log(1/\delta_1)) + 8\alpha\sqrt{Mtk(Mt + \log(2Mt^2/\delta_2))} \tag{25}$$

Let $\delta_1 = \delta/|\Phi_\alpha^2|, \delta_2 = \delta$, and notice $\log|\Phi_\alpha^2| \le 2\log(\mathcal{N}(\Phi, \alpha, \|\cdot\|_\infty))$, we conclude that with probability at least $1 - 2\delta$, for every $t \in \mathbb{N}$

$$\left\|\hat{f}_t - f_\theta\right\|_{2,E_t}^2 \le 12Mk + 12\log(\mathcal{N}(\Phi, \alpha, \|\cdot\|_\infty)/\delta) + 8\alpha\sqrt{Mtk(Mt + \log(2Mt^2/\delta))} \tag{26}$$

where the right handside is exactly our defined $\beta_t(\Phi, \alpha, \delta)$, hence our conclusion holds. $\qquad\square$

**Lemma 2.** *If $(\beta_t \ge 0 \mid t \in \mathbb{N})$ is a nondecreasing sequence and $\mathcal{F}_t := \left\{ f \in \mathcal{F}^{\otimes M} : \|f - \hat{f}_t^{LS}\|_{2,E_t} \le \sqrt{\beta_t} \right\}$. Also, denote $\mathcal{F} = \mathcal{L} \circ \Phi : \mathcal{C} \times \mathcal{A} \mapsto [0,1]$, we have*

$$\sum_{t=1}^{T} \mathbb{I}(w_{\mathcal{F}_t}(\boldsymbol{X}_t) > \epsilon) \le \left(\frac{4M\beta_T}{\epsilon^2} + 1\right)\dim_E(\mathcal{F}, \epsilon)$$

*Proof.* The main structure of this proof is similar to proposition 3, section C in Eluder dimension's paper, and we will only point out the subtle details that makes the difference. We will show that if $w_{\mathcal{F}_t}(\boldsymbol{X}_t) > \epsilon$, then $\boldsymbol{X}_t$ is $\epsilon$-dependent on fewer than $4M\beta_T/\epsilon^2$ disjoint subsequences of $(\boldsymbol{X}_1, \ldots, \boldsymbol{X}_{t-1})$. Note that if $w_{\mathcal{F}_t}(\boldsymbol{X}_t) > \epsilon$, there are $\overline{f}, \underline{f} \in \mathcal{F}_t$ such that $\sum_{i=1}^{M} \overline{f}^{(i)}(\boldsymbol{x}_{t,i}) - \underline{f}^{(i)}(\boldsymbol{x}_{t,i}) > \epsilon$. By definition, if $\boldsymbol{X}_t$ is $\epsilon$-dependent on a subsequence $(\boldsymbol{X}_{t_1}, \boldsymbol{X}_{t_2}, \ldots, \boldsymbol{X}_{t_k})$ of $(\boldsymbol{X}_1, \ldots, \boldsymbol{X}_{t-1})$, then we know

$$\sum_{j=1}^{k}\left(\sum_{i=1}^{M} \overline{f}^{(i)}(\boldsymbol{x}_{t_j,i}) - \underline{f}^{(i)}(\boldsymbol{x}_{t_j,i})\right)^2 > \epsilon^2$$

It follows that, if $\boldsymbol{X}_t$ is $\epsilon$-dependent on $K$ disjoint subsequences of $(\boldsymbol{X}_1, \ldots, \boldsymbol{X}_{t-1})$, then

$$\|\overline{f} - \underline{f}\|_{2,E_t}^2 = \sum_{s=1}^{t} \sum_{i=1}^{M} \left( \overline{f}^{(i)}(\boldsymbol{x}_{s,i}) - \underline{f}^{(i)}(\boldsymbol{x}_{s,i}) \right)^2 \tag{27}$$

$$\geq \frac{1}{M} \sum_{s=1}^{t} \left( \sum_{i=1}^{M} \overline{f}^{(i)}(\boldsymbol{x}_{s,i}) - \underline{f}^{(i)}(\boldsymbol{x}_{s,i}) \right)^2 \qquad \text{(Cauchy Inequality)}$$

$$> \frac{K\epsilon^2}{M} \tag{28}$$

By triangle inequality we have

$$\|\overline{f} - \underline{f}\|_{2,E_t} \leq \|\overline{f} - \hat{f}_t^{LS}\|_{2,E_t} + \|\hat{f}_t^{LS} - \underline{f}\|_{2,E_t} \leq 2\sqrt{\beta_t} \leq 2\sqrt{\beta_T} \tag{29}$$

and it follows that $K < 4M\beta_T/\epsilon^2$.

Notice that essentially we are analyzing scalar output function $g(\boldsymbol{X}_t) = \sum_{i=1}^{M} f^{(i)}(\boldsymbol{x}_{t,i})$ where $f \in \mathcal{F}^{\otimes M}$. Hence if we denote any $f \in \mathcal{F}^{\otimes M}$ as $f(\cdot) = \phi(\cdot)^\top \boldsymbol{\Theta}$, then $g(\cdot) = \phi(\cdot)^\top \boldsymbol{w} \in \mathcal{F}, \boldsymbol{w} = \boldsymbol{\Theta} \cdot \mathbf{1}$. Hence from original eluder dimension paper we know in any action sequence $(\boldsymbol{X}_1, \ldots, \boldsymbol{X}_\tau)$, there must exist some element $\boldsymbol{X}_j$ that is $\epsilon$-dependent on at least $\tau/d - 1$ disjoint subsequences of $(\boldsymbol{X}_1, \ldots, \boldsymbol{X}_\tau)$, where $d := \dim_E(\mathcal{F}, \epsilon)$. Finally we select $\boldsymbol{X}_1, \ldots, \boldsymbol{X}_\tau$ as those actions that $w_{\mathcal{F}_t} > \epsilon$, combine these two facts above and get $\tau/d - 1 \leq 4M\beta_T/\epsilon^2$. Hence $\tau \leq (4M\beta_T/\epsilon^2 + 1)d$, which is our desired conclusion.

## B    Linear MDP Regret Analysis

Apart from the notations section 3, we add more symbols for the regret analysis. We use $Q[f]$ or $Q[\phi \circ \boldsymbol{\theta}]$ to denote the Q-value function parametrized by function $f$ as $Q[f](s,a) = f(s,a)$ or $Q[\phi \circ \boldsymbol{\theta}](s,a) = \phi(s,a)^\top \boldsymbol{\theta}$ (similar for $V[f]$ as state's value estimation function). Also, based on assumption 2.1, for any $\left\{ Q_{h+1}^{(i)} \right\}_{i=1}^{M}$, there always exists $\dot{f}_h [Q_{h+1}] \in \mathcal{F}^{\otimes M}$ such that

$$\Delta_h^{(i)} \left( Q_{h+1}^{(i)} \right)(s,a) = \mathcal{T}_h^i \left( Q_{h+1}^{(i)} \right)(s,a) - \dot{f}_h^{(i)}(s,a) \tag{30}$$

where the approximation error $\left\| \Delta_h^{(i)} \left( Q_{h+1}^{(i)} \right) \right\| \leq \mathcal{I}$ for $\forall\, i \in [M]$. Here $\dot{f}_h[Q_{h+1}]$ indicates that function $\dot{f}_h$ has dependence on Q-value function $Q_{h+1}$ on next level $h+1$. In following analysis, we will use different annotations for different function approximation as below

- $f_h^{(i)*}(\cdot, \cdot) = \phi^*(\cdot, \cdot)^\top \boldsymbol{\theta}_h^{(i)*}$ is the "best" Q-value function approximation in $\mathcal{Q}_h$ for task $i$ at level $h$.

- $\hat{f}_h^{(i)}(\cdot, \cdot) = \hat{\phi}(\cdot, \cdot)^\top \hat{\boldsymbol{\theta}}_i$ is the empirical least-square minimizer solution for task $i$ at level $h$.

- $\dot{f}_h^{(i)}(\cdot, \cdot) = \dot{\phi}(\cdot, \cdot)^\top \dot{\boldsymbol{\theta}}_i$ is the value approximation function $\mathcal{T}_h^{(i)} Q_{h+1}^{(i)}$ induced by $Q_{h+1}^{(i)}$ for task $i$ at level $h$.

- $\tilde{f}_h^{(i)}(\cdot, \cdot) = \tilde{\phi}(\cdot, \cdot)^\top \tilde{\boldsymbol{\theta}}_i$ is the optimism Q-value approximation function for task $i$ at level $h$.

- $\bar{f}_h^{(i)}(\cdot, \cdot) = \bar{\phi}(\cdot, \cdot)^\top \bar{\boldsymbol{\theta}}_i$ is the nearest neighbor in covering set for task $i$ at level $h$.

### B.1    Main Proof sketch

The overall structure is similar to bandits, the main difference here is that we need to take care of the transition dynamics.

Firstly, we decompose the total regret into following terms

$$\text{Reg}(T) = \sum_{t=1}^{T} \sum_{i=1}^{M} \left( V_1^{(i)\star} - V_1^{\pi_t^i} \right) \left( s_{1,t}^{(i)} \right) \tag{31}$$

$$= \sum_{t=1}^{T} \sum_{i=1}^{M} \left( V_1^{(i)\star} - V_1^{(i)} \left[ \tilde{f}_{1,t}^{(i)} \right] \right) \left( s_{1,t}^{(i)} \right) + \sum_{t=1}^{T} \sum_{i=1}^{M} \left( V_1^{(i)} \left[ \tilde{f}_{1,t}^{(i)} \right] - V_1^{\pi_t^i} \right) \left( s_{1,t}^{(i)} \right) \tag{32}$$

$$\leq \sum_{t=1}^{T} \sum_{i=1}^{M} \left( V_1^{(i)} \left[ \tilde{f}_{1,t}^{(i)} \right] - V_1^{\pi_t^i} \right) \left( s_{1,t}^{(i)} \right) + MHT\mathcal{I}. \tag{33}$$

The inequality is because according to lemma 3, we have at each episode $t \in [T]$

$$\sum_{i=1}^{M} \left( V_1^{i\star} - V_1^{(i)} \left[ \tilde{f}_{1,t}^{(i)} \right] \right) \left( s_{1,t}^{(i)} \right) \leq MH\mathcal{I}$$

$$\implies \sum_{t=1}^{T} \sum_{i=1}^{M} \left( V_1^{i\star} - V_1^{(i)} \left[ \tilde{f}_{1,t}^{(i)} \right] \right) \left( s_{1,t}^{(i)} \right) \leq MHT\mathcal{I}.$$

Denote $a_{h,t}^{(i)} = \pi_t^i \left( s_{ht}^{(i)} \right)$, $Q_h^{(i)}[\tilde{f}_{h,t}^{(i)}] = \tilde{Q}_{h,t}^{(i)}$ and $V_h^{(i)}[\tilde{f}_{h,t}^{(i)}] = \tilde{V}_{h,t}^{(i)}$ for short. We have for any $t \in [T], h \in [H]$

$$\sum_{i=1}^{M} \left( \tilde{V}_{h,t}^{(i)} - V_{h,t}^{\pi_t^i} \right) \left( s_{h,t}^{(i)} \right) = \sum_{i=1}^{M} \left( \tilde{Q}_{h,t}^{(i)} - Q_{h,t}^{\pi_t^i} \right) \left( s_{h,t}^{(i)}, a_{h,t}^{(i)} \right) \tag{34}$$

$$= \sum_{i=1}^{M} \left( \tilde{Q}_{h,t}^{(i)} - \mathcal{T}_h^{(i)} \tilde{Q}_{h+1,t}^{(i)} \right) \left( s_{1,t}^{(i)}, a_{h,t}^{(i)} \right) + \sum_{i=1}^{M} \left( \mathcal{T}_h^{(i)} \tilde{Q}_{h+1,t}^{(i)} - Q_{h,t}^{\pi_t^i} \right) \left( s_{h,t}^{(i)}, a_{h,t}^{(i)} \right) \tag{35}$$

Since the failure event $\bigcup_{t=1}^{T} \bigcup_{h=1}^{H} E_{ht}$ only happens with probability $\delta$ according to lemma 6, and the addition of regret when it happens is constant bounded, we will simply assume that it does not happen. Then applying lemma 5, we have

$$\sum_{i=1}^{M} \left( \tilde{Q}_{h,t}^{(i)} - \mathcal{T}_h^{(i)} \tilde{Q}_{h+1,t}^{(i)} \right) \left( s_{h,t}^{(i)}, a_{h,t}^{(i)} \right) \leq M\mathcal{I} + 2w_{\mathcal{F}_{h,t}} \left( \boldsymbol{x}_{h,t} \right). \tag{36}$$

where $\boldsymbol{x}_{h,t} = \left[ (s_{h,t}^{(1)}, a_{h,t}^{(1)}), \ldots, (s_{h,t}^{(M)}, a_{h,t}^{(M)}) \right]$ denotes the stacked input for all state-action pair at level $h$, episode $t$.

Next, we expand the second summation in (35) and have

$$\sum_{i=1}^{M} \left( \mathcal{T}_h^{(i)} \tilde{Q}_{h+1,t}^{(i)} - Q_{h,t}^{\pi_t^i} \right) \left( s_{h,t}^{(i)}, a_{h,t}^{(i)} \right) = \sum_{i=1}^{M} \mathbb{E}_{s' \sim \mathcal{P}_h^{(i)} \left( \cdot | s_{h,t}^{(i)}, a_{h,t}^{(i)} \right)} \left[ \left( \tilde{V}_{h+1,t}^{(i)} - V_{h+1}^{\pi_t^i} \right) (s') \right] \tag{37}$$

$$= \sum_{i=1}^{M} \left( \tilde{V}_{h+1,t}^{(i)} - V_{h+1}^{\pi_t^i} \right) \left( s_{h+1,t}^{(i)} \right) + \sum_{i=1}^{M} \varsigma_{h,t}^{(i)} \tag{38}$$

where $\varsigma_{h,t}^{(i)}$ is a martingale difference with respect to history $\mathcal{H}_{h,t}$ defined by

$$\varsigma_{h,t}^{(i)} \stackrel{\text{def}}{=} \mathbb{E}_{s' \sim \mathcal{P}_h^{(i)} \left( \cdot | s_{h,t}^{(i)}, a_{h,t}^{(i)} \right)} \left[ \left( \tilde{V}_{h+1,t}^{(i)} - V_{h+1}^{\pi_t^i} \right) (s') \right] - \left( \tilde{V}_{h+1,t}^{(i)} - V_{h+1}^{\pi_t^i} \right) (s') \tag{39}$$

According to assumption 2.2 we know that $|\varsigma_{h,t}^{(i)}| \leq 4$, hence by Azuma-Hoeffding's inequality, we know that with probability at least $1 - \delta/2$, for any $t \in [T]$ and $i \in [M]$

$$\sum_{j=1}^{t} \varsigma_{h,t}^{(i)} \leq 4\sqrt{2t \log \frac{2T}{\delta}}. \tag{40}$$

We can then apply (38) recursively from $h = 1$ to $H$, which gives

$$\text{Reg}(T) \leq \sum_{t=1}^{T} \sum_{i=1}^{M} \left( \tilde{V}_{1,t}^{(i)} - V_1^{\pi^i} \right) \left( s_{1,t}^{(i)} \right) + MHT\mathcal{I} \tag{41}$$

$$\leq 2MHT\mathcal{I} + \sum_{t=1}^{T} \sum_{h=1}^{H} 2w_{\mathcal{F}_t}(\boldsymbol{x}_{h,t}) + \sum_{i=1}^{M} \sum_{h=1}^{H} \sum_{t=1}^{T} \zeta_{h,t}^{(i)} \tag{42}$$

According to lemma 2 we know that

$$\sum_{t=1}^{T} w_{\mathcal{F}_t}(\boldsymbol{x}_{h,t}) \leq \left( \frac{4M\beta_{h,T}}{\alpha^2} + 1 \right) \dim_E(\mathcal{F}, \alpha) \tag{43}$$

where $\beta_{h,t} = \tilde{O}(Mk + \log \mathcal{N}(\Phi, \alpha, \|\cdot\|_\infty) + MT\mathcal{I}^2)$. Summarizing all inequality above and we have the final regret bound as

$$\text{Reg}(T) = 2MHT\mathcal{I} + \sum_{t=1}^{T} \sum_{h=1}^{H} 2w_{\mathcal{F}_t}(\boldsymbol{x}_{h,t}) + \sum_{i=1}^{M} \sum_{h=1}^{H} \sum_{t=1}^{T} \zeta_{h,t}^{(i)} \tag{44}$$

$$= \tilde{O}\left( MHT\mathcal{I} + \tilde{O}(\sqrt{Mk + \log \mathcal{N}(\Phi, \alpha, \|\cdot\|_\infty) + MT\mathcal{I}^2})H\sqrt{MT \dim_E(\mathcal{F}, \alpha)} + MH\sqrt{T} \right) \tag{45}$$

Set $\alpha = \frac{1}{kMT}$, we have the regret bound as

$$\tilde{O}\left( H\sqrt{\dim_E(\mathcal{F}, (kMT)^{-1})} \left( M\sqrt{Tk} + \sqrt{MT \log \mathcal{N}(\Phi, (kMT)^{-1}, \|\cdot\|_\infty)} + MT\mathcal{I} \right) \right).$$

## B.2 Detailed Lemma Proof

**Lemma 3.** *Let $V_1^{i\star}$ be the value of optimal policy and $V_1^i \left[ \tilde{f}_{1,t}^{(i)} \right]$ be the optimistic value estimation defined in main proof. We have the accuracy guarantee as*

$$\sum_{i=1}^{M} \left( V_1^{(i)\star} - V_1^{(i)} \left[ \tilde{f}_{1,t}^{(i)} \right] \right) \left( s_{1,t}^{(i)} \right) \leq MH\mathcal{I}. \tag{46}$$

*Proof.* Recursively define the closest value approximator function $f_h^* = (\phi_h^*)^\top \Theta_h^*$ at level $h$ within function class $\mathcal{F}^{\otimes M}$ as

$$\phi_h^*, \Theta_h^* \overset{\text{def}}{=} \underset{\phi \in \Phi, \Theta = [\boldsymbol{\theta}_1, \ldots, \boldsymbol{\theta}_M] \in \mathbb{R}^{k \times M}}{\arg\min} \underset{s,a,i}{\sup} \left| \phi(s,a)^\top \boldsymbol{\theta}_h^{(i)} - \mathcal{T}_h^{(i)} Q_{h+1}^{(i)} \left[ \phi_{h+1}^* \circ \boldsymbol{\theta}_{h+1}^{(i)*} \right] (s,a) \right| \tag{47}$$

with $\boldsymbol{\theta}_{H+1}^{(i)} = \mathbf{0}$ for any $i \in [M]$ and $\Theta_h^* = \left[ \boldsymbol{\theta}_h^{(1)*}, \ldots, \boldsymbol{\theta}_h^{(M)*} \right]$. By lemma 6 in [?] we have

$$\underset{(s,a) \in \mathcal{S} \times \mathcal{A}, i \in [M]}{\sup} \left| Q_h^{(i)\star}(s,a) - \phi_h^*(s,a)^\top \boldsymbol{\theta}_h^{(i)*} \right| \leq (H - h + 1)\mathcal{I}. \tag{48}$$

where $Q_h^{(i)\star}$ is the optimal value function for task $i$.

Next, we will show that $f_h^*$ is a feasible solution for the optimization of $\mathcal{F}_t$. This is achieved via inductive construction. For $h = H + 1$ we know it holds trivially because $\tilde{f}_{H+1}^{(i)} = f_{H+1}^{(i)*} = \mathbf{0}$. Now we suppose that $\beta_{h,t}$ for $k = h + 1, \ldots, H$ satisfies that we can always find $\tilde{f}_k^{(i)} = f_k^{(i)*}$. Then from the definition of $f_h^{(i)*}$ we can always properly set $\mathcal{F}_{h,t}$ (to be specified later) to let it contain

$$\dot{f}_h^{(i)} \left[ V_{h+1}^{(i)} \left[ f_{h+1}^{(i)*} \right] \right] = f_h^{(i)*}. \tag{49}$$

By lemma 4, we have

$$\left\| \hat{f}_h \left[ V_{h+1} \left[ f_{h+1}^* \right] \right] - \dot{f}_h \left[ V_{h+1} \left[ f_{h+1}^* \right] \right] \right\|_{2,E_t}^2 \leq \beta_{h,t}. \tag{50}$$

Therefore, set $\beta_{h,t}$ as the function we set *does* let $f_h^{(i)*} \in \mathcal{F}_{h,t}$.

Finally, we can finish the proof from showing that

$$\sum_{i=1}^{M} V_1^{(i)} \left[ \tilde{f}_{1,t}^{(i)} \right] \left( s_{1,t}^{(i)} \right) \tag{51}$$

$$= \sum_{i=1}^{M} \max_{a \in \mathcal{A}} \tilde{f}_{1,t}^{(i)} \left( s_{1,t}^{(i)}, a \right) \tag{52}$$

$$\geq \sum_{i=1}^{M} \max_{a \in \mathcal{A}} f_{1,t}^{(i)*} \left( s_{1,t}^{(i)}, a \right) \qquad \text{(because } f_1^{(i)*} \in \mathcal{F}_t) \tag{53}$$

$$\geq \sum_{i=1}^{M} f_{1,t}^{(i)*} \left( s_{1,t}^{(i)}, \pi_1^{i\star} \left( s_{1,t}^{(i)} \right) \right) \tag{53}$$

$$\geq \sum_{i=1}^{M} Q_1^{(i)\star} \left( s_{1,t}^{(i)}, \pi_1^{i\star} \left( s_{1,t}^{(i)} \right) \right) - MH\mathcal{I} \qquad \text{(By (48))}$$

$$\geq \sum_{i=1}^{M} V_1^{(i)\star} \left( s_{1,t}^{(i)} \right) - MH\mathcal{I}. \tag{54}$$

$\square$

**Lemma 4.** *For any episode $t \in [T]$, level $h \in [H]$ and any Q-value function at next level $\{Q_{h+1}^{(i)}\}_{i=1}^{M} \in \mathcal{Q}_{h+1}$, denote $\dot{f}_{h,t}$ as the best fit Q-value estimation induced by $Q_{h+1}^{(i)}$ minimizing Bellman error, we have*

$$\left\| \hat{f}_{h,t}[Q_{h+1}] - \dot{f}_{h,t}[Q_{h+1}] \right\|_{2,E_t}^2 \leq \beta_{h,t} \stackrel{\text{def}}{=} \left( B_{h,1} + \sqrt{MT\mathcal{I}} + \sqrt{B_{h,2}} \right)^2. \tag{55}$$

*The $B_{h,1}$ and $B_{h,2}$ are from Lemma 6. Equivalently saying, this means that $\dot{f}_{h,t}$ is contained in set $\mathcal{F}_{h,t}$ defined as*

$$\mathcal{F}_{h,t} \stackrel{\text{def}}{=} \left\{ f \in \mathcal{F}^{\otimes M} : \left\| f - \hat{f}_{h,t}[Q_{h+1}] \right\|_{2,E_t}^2 \leq \beta_{h,t} \right\}.$$

*Proof.* By the empirical optimality of $\hat{f}_{h,t}$, we know

$$\sum_{i=1}^{M} \left\| \hat{f}_{h,t}^{(i)}(\boldsymbol{X}_{h,t}) - \boldsymbol{y}_{h,t}^{(i)} \right\|^2 \leq \sum_{i=1}^{M} \left\| \dot{f}_{h,t}^{(i)}(\boldsymbol{X}_{h,t}) - \boldsymbol{y}_{h,t}^{(i)} \right\|^2. \tag{56}$$

Here we abuse the notation and use $\hat{f}_{h,t}^{(i)}(\boldsymbol{X}_{h,t})$ to denote function $\hat{f}_{h,t}^{(i)}$'s output on all the state-action pair $\boldsymbol{X}_{h,t}$ in the first $t-1$ episodes at level $h$ for task $i$, also $\boldsymbol{y}_{h,t}^{(i)}$ is the corresponding target value label. This inequality implies that

$$\sum_{i=1}^{M} \left\| \hat{f}_{h,t}^{(i)}(\boldsymbol{X}_{h,t}) - \dot{f}_{h,t}^{(i)}(\boldsymbol{X}_{h,t}) \right\|^2 \tag{57}$$

$$\leq 2 \sum_{i=1}^{M} \left\langle \boldsymbol{\Delta}_{h,t}^{(i)}, \hat{f}_{h,t}^{(i)}(\boldsymbol{X}_{h,t}) - \dot{f}_{h,t}^{(i)}(\boldsymbol{X}_{h,t}) \right\rangle + 2 \sum_{i=1}^{M} \left\langle \boldsymbol{z}_{h,t}^{(i)}, \hat{f}_{h,t}^{(i)}(\boldsymbol{X}_{h,t}) - \dot{f}_{h,t}^{(i)}(\boldsymbol{X}_{h,t}) \right\rangle \tag{58}$$

where

$$\boldsymbol{\Delta}_{h,t}^{(i)} \stackrel{\text{def}}{=} \left[ \Delta_{h,1}^{(i)}(Q_{h+1}^{(i)})(s_{h,1}^{(i)}, a_{h,2}^{(i)}) \quad \Delta_{h,2}^{(i)}(Q_{h+1}^{(i)})(s_{h,2}^{(i)}, a_{h,2}^{(i)}) \quad \cdots \quad \Delta_{h,t-1}^{(i)}(Q_{h+1}^{(i)})(s_{h,t-1}^{(i)}, a_{h,t-1}^{(i)}) \right]$$

is the Bellman error for Q-value approximation, each $\Delta_{h,j}^{(i)}(Q_{h+1}^{(i)})(s_{h,j}^{(i)}, a_{h,j}^{(i)})$ is defined in (30). And

$$\boldsymbol{z}_{h,t}^{(i)} \stackrel{\text{def}}{=} \left[ z_{h,1}^{(i)}(Q_{h+1}^{(i)})(s_{h,1}^{(i)}, a_{h,2}^{(i)}) \quad \cdots \quad z_{h,t-1}^{(i)}(Q_{h+1}^{(i)})(s_{h,t-1}^{(i)}, a_{h,t-1}^{(i)}) \right]$$

where $z_{h,j}^{(i)} \left( Q_{h+1}^{(i)} \right) \left( s_{h,j}^{(i)}, a_{h,j}^{(i)} \right) \overset{\text{def}}{=} R \left( s_{h,j}^{(i)}, a_{h,j}^{(i)} \right) + \max_{a \in \mathcal{A}} Q_{h+1}^{(i)} \left( s_{h+1,j}^{(i)}, a \right) -$ $\mathcal{T}_h^{(i)} \left( Q_{h+1}^{(i)} \right) \left( s_{h,j}^{(i)}, a_{h,j}^{(i)} \right)$ is the finite sampling noise.

Next, we are going to bound the two terms in (58). For the first term, we have

$$\sum_{i=1}^{M} \left\langle \mathbf{\Delta}_{h,t}^{(i)}, \hat{f}_{h,t}^{(i)}(\mathbf{X}_{h,t}) - \dot{f}_{h,t}^{(i)}(\mathbf{X}_{h,t}) \right\rangle \tag{59}$$

$$\leq \sum_{i=1}^{M} \left\| \mathbf{\Delta}_{h,t}^{(i)} \right\| \cdot \left\| \hat{f}_{h,t}^{(i)}(\mathbf{X}_{h,t}) - \dot{f}_{h,t}^{(i)}(\mathbf{X}_{h,t}) \right\| \tag{60}$$

$$\leq \sqrt{T\mathcal{I}} \cdot \sum_{i=1}^{M} \left\| \hat{f}_{h,t}^{(i)}(\mathbf{X}_{h,t}) - \dot{f}_{h,t}^{(i)}(\mathbf{X}_{h,t}) \right\| \tag{61}$$

$$\leq \sqrt{MT\mathcal{I}} \cdot \left\| \hat{f}_{h,t} - \dot{f}_{h,t} \right\|_{2,E_t} \tag{62}$$

By lemma 6, when the failure case does not happen, we have

$$\sum_{i=1}^{M} \left\langle z_{h,t}^{(i)}, \hat{f}_{h,t}^{(i)}(\mathbf{X}_{h,t}) - \dot{f}_{h,t}^{(i)}(\mathbf{X}_{h,t}) \right\rangle \leq B_{h,1} \cdot \left\| \hat{f}_{h,t} - \dot{f}_{h,t} \right\|_{2,E_t} + B_{h,2} \tag{63}$$

where

$$B_{h,1} = \sqrt{2Mk + \log(\mathcal{N}(\Phi, (kMT)^{-1}, \|\cdot\|_\infty)/\delta)} + 1 \tag{64}$$

$$B_{h,2} = 2\sqrt{MT + \log(2MT^2/\delta)} \tag{65}$$

Adding the bound for two terms and we get

$$\left\| \hat{f}_{h,t} - \dot{f}_{h,t} \right\|_{2,E_t}^2 \leq (B_{h,1} + \sqrt{MT\mathcal{I}}) \cdot \left\| \hat{f}_{h,t} - \dot{f}_{h,t} \right\|_{2,E_t} + B_{h,2} \tag{66}$$

$$\implies \left\| \hat{f}_{h,t} - \dot{f}_{h,t} \right\|_{2,E_t}^2 \leq \left( B_{h,1} + \sqrt{MT\mathcal{I}} + \sqrt{B_{h,2}} \right)^2 \overset{\text{def}}{=} \beta_{h,t} \tag{67}$$

which completes the proof. $\qquad\square$

**Lemma 5.** *If the failure event in lemma 6 does not happen, for any feasible solution $Q_h^{(i)} \left[ \tilde{f}_h^{(i)} \right]$ in the definition of $\mathcal{F}_{h,t}$, and any $h \in [H]$, $t \in [T]$, we have*

$$\sum_{i=1}^{M} \left| \left( \tilde{Q}_{h,t}^{(i)} - \mathcal{T}_h^{(i)} \tilde{Q}_{h+1,t}^{(i)} \right) \left( s_{h,t}^{(i)}, a_{h,t}^{(i)} \right) \right| \leq M\mathcal{I} + 2w_{\mathcal{F}_{h,t}} \left( \mathbf{x}_{h,t} \right), \tag{68}$$

*where $\mathbf{x}_{h,t} = \left[ (s_{h,t}^{(1)}, a_{h,t}^{(1)}), \dots, (s_{h,t}^{(M)}, a_{h,t}^{(M)}) \right]$ denotes the stacked input for all state-action pair at level $h$, episode $t$.*

*Proof.*

$$\sum_{i=1}^{M} \left| \left( \tilde{Q}_{h,t}^{(i)} - \mathcal{T}_h^{(i)} \tilde{Q}_{h+1,t}^{(i)} \right) \left( s_{h,t}^{(i)}, a_{h,t}^{(i)} \right) \right| \tag{69}$$

$$= \sum_{i=1}^{M} \left| \tilde{Q}_{h,t}^{(i)}(s, a) - \dot{f}_h^{(i)} \left[ \tilde{Q}_{h+1}^{(i)} \right] \left( s_{h,t}^{(i)}, a_{h,t}^{(i)} \right) - \Delta_h^{(i)} \left( \tilde{Q}_{h+1}^{(i)} \right) \left( s_{h,t}^{(i)}, a_{h,t}^{(i)} \right) \right| \tag{70}$$

$$\leq M\mathcal{I} + \sum_{i=1}^{M} \left| \tilde{f}_{h,t}^{(i)} \left( s_{h,t}^{(i)}, a_{h,t}^{(i)} \right) - \dot{f}_h^{(i)} \left[ \tilde{Q}_{h+1}^{(i)} \right] \left( s_{h,t}^{(i)}, a_{h,t}^{(i)} \right) \right| \tag{71}$$

$$\leq M\mathcal{I} + \sum_{i=1}^{M} \left| \tilde{f}_{h,t}^{(i)} \left( s_{h,t}^{(i)}, a_{h,t}^{(i)} \right) - \hat{f}_h^{(i)} \left( s_{h,t}^{(i)}, a_{h,t}^{(i)} \right) \right| + \left| \hat{f}_h^{(i)} \left( s_{h,t}^{(i)}, a_{h,t}^{(i)} \right) - \dot{f}_h^{(i)} \left[ \tilde{Q}_{h+1}^{(i)} \right] \left( s_{h,t}^{(i)}, a_{h,t}^{(i)} \right) \right| \tag{72}$$

According to our construction, we know that both $\tilde{f}_{h,t}^{(i)}$ and $\dot{f}_h^{(i)}$ are contained in $\mathcal{F}_{h,t}$, therefore we have $\sum_{i=1}^{M} \left| \tilde{f}_{h,t}^{(i)} \left( s_{h,t}^{(i)}, a_{h,t}^{(i)} \right) - \hat{f}_h^{(i)} \left( s_{h,t}^{(i)}, a_{h,t}^{(i)} \right) \right| \leq w_{\mathcal{F}_{h,t}} \left( \boldsymbol{x}_{h,t} \right)$ and $\sum_{i=1}^{M} \left| \dot{f}_{h,t}^{(i)} \left[ \tilde{Q}_{h+1}^{(i)} \right] \left( s_{h,t}^{(i)}, a_{h,t}^{(i)} \right) - \hat{f}_h^{(i)} \left( s_{h,t}^{(i)}, a_{h,t}^{(i)} \right) \right| \leq w_{\mathcal{F}_{h,t}} \left( \boldsymbol{x}_{h,t} \right)$, where $\boldsymbol{x}_{h,t} = \left[ (s_{h,t}^{(1)}, a_{h,t}^{(1)}), \ldots, (s_{h,t}^{(M)}, a_{h,t}^{(M)}) \right]$ denotes the stacked input for all state-action pair at level $h$, episode $t$.

Summarizing all the inequalities and we know the whole lemma holds. $\qquad \square$

**Lemma 6.** (Probability bound for failure event) *In this lemma we denote $\hat{f}_h^{(i)} \left[ Q_{h+1}^{(i)} \right]$ as $\hat{f}_h^{(i)}$ for the sake of simplicity (similar for $\dot{f}_h^{(i)}$). Define event $E_{h,t}$ as*

$$E_{h,t} \overset{\text{def}}{=} \mathbb{I} \left[ \exists \{Q_{h+1}^{(i)}\}_{i=1}^{M} \quad \sum_{i=1}^{M} \left\langle \boldsymbol{z}_{h,t}^{(i)}, \hat{f}_h^{(i)}(\boldsymbol{X}_{h,t}) - \dot{f}_h^{(i)}(\boldsymbol{X}_{h,t}) \right\rangle > B_{h,1} \cdot \left\| \hat{f}_h^{(i)} - \dot{f}_h^{(i)} \right\|_{2,E_t} + B_{h,2} \right]$$

(73)

*where $B_{h,1}$ and $B_{h,2}$ will be specified later. We have*

$$\mathbb{P} \left( \bigcup_{t=1}^{T} \bigcup_{h=1}^{H} E_{h,t} \right) \leq \delta. \tag{74}$$

*Proof.* Similar to lemma 1, we can find a $\alpha$-cover $\Phi_\alpha$ for $\Phi$ such that for any Q-value function $\left( Q_{h+1}^{(1)}[\phi \circ \boldsymbol{\theta}_1], Q_{h+1}^{(2)}[\phi \circ \boldsymbol{\theta}_2], \ldots, Q_{h+1}^{(M)}[\phi \circ \boldsymbol{\theta}_M] \right)$, we can find $\bar{\phi} \in \Phi_\alpha$ and $\bar{\boldsymbol{\theta}}_i$ for $i \in [M]$ such that for any $(s,a) \in \mathcal{S} \times \mathcal{A}$ and any $i \in [M]$

$$\left| Q_{h+1}^{(i)}(s,a) - \bar{\phi}(s,a)^\top \bar{\boldsymbol{\theta}}_i \right| \leq \sqrt{k}\alpha. \tag{75}$$

Define $\bar{Q}_{h+1}^{(i)} = Q_{h+1}^{(i)} \left[ \bar{\phi} \circ \boldsymbol{\theta}_i \right]$ and further let

$$\bar{\boldsymbol{z}}_{h,t}^{(i)} \overset{\text{def}}{=} \left[ z_{h,1}^{(i)} \left( \bar{Q}_{h+1}^{(i)} \right) \left( s_{h,1}^{(i)}, a_{h,1}^{(i)} \right) \quad \cdots \quad z_{h,t-1}^{(i)} \left( \bar{Q}_{h+1}^{(i)} \right) \left( s_{h,t-1}^{(i)}, a_{h,t-1}^{(i)} \right) \right] \in \mathbb{R}^{t-1}$$

then we have

$$\sum_{i=1}^{M} \left\langle \boldsymbol{z}_{h,t}^{(i)}, \hat{f}_h^{(i)}(\boldsymbol{X}_{h,t}) - \dot{f}_h^{(i)}(\boldsymbol{X}_{h,t}) \right\rangle \tag{76}$$

$$= \sum_{i=1}^{M} \left\langle \bar{\boldsymbol{z}}_{h,t}^{(i)}, \hat{f}_h^{(i)}(\boldsymbol{X}_{h,t}) - \dot{f}_h^{(i)}(\boldsymbol{X}_{h,t}) \right\rangle \tag{77}$$

$$+ \sum_{i=1}^{M} \left\langle \boldsymbol{z}_{h,t}^{(i)} - \bar{\boldsymbol{z}}_{h,t}^{(i)}, \hat{f}_h^{(i)}(\boldsymbol{X}_{h,t}) - \dot{f}_h^{(i)}(\boldsymbol{X}_{h,t}) \right\rangle \tag{78}$$

$$\tag{79}$$

Notice that for fixed $\bar{f}_h^{(i)}(\cdot,\cdot) = \phi(\cdot,\cdot)^\top \bar{\boldsymbol{\theta}}_{h+1}^{(i)}$, each $z_{h,1}^{(i)} \left( \bar{Q}_{h+1}^{(i)} \right) \left( s_{h,1}^{(i)}, a_{h,2}^{(i)} \right)$ is a zero-mean 1-sub-Gaussian random variable conditioned on past history. Therefore we can treat it as $\eta_{t,i} = z_{h,t}^{(i)}$ in Lemma 1 and get

$$\sum_{i=1}^{M} \left\langle \bar{\boldsymbol{z}}_{h,t}^{(i)}, \hat{f}_h^{(i)}(\boldsymbol{X}_{h,t}) - \dot{f}_h^{(i)}(\boldsymbol{X}_{h,t}) \right\rangle \tag{80}$$

$$\leq \sqrt{2Mk + \log(1/\delta_1)} \left\| \hat{f}_{h,t} - \dot{f}_{h,t} \right\|_{2,E_t} + 2\alpha \sqrt{Mtk(Mt + \log(2Mt^2/\delta_2))}. \tag{81}$$

Setting $\delta_1 = \frac{\delta}{2|\Phi^\alpha|}, \delta_2 = \delta/2$ and get

$$\sum_{i=1}^{M} \left\langle \bar{z}_{h,t}^{(i)}, \hat{f}_h^{(i)}(\boldsymbol{X}_{h,t}) - \dot{f}_h^{(i)}(\boldsymbol{X}_{h,t}) \right\rangle \tag{82}$$

$$\leq \sqrt{2Mk + \log(\mathcal{N}(\Phi, \alpha, \|\cdot\|_\infty)/\delta)} \cdot \left\|\hat{f}_{h,t} - \dot{f}_{h,t}\right\|_{2, E_t} + 2\alpha\sqrt{MTk(MT + \log(2MT^2/\delta))}. \tag{83}$$

By union bound, we know it holds for any $\bar{f}_h$ with probability at least $1 - |\Phi^\alpha|\delta_1 = 1 - \delta$. Also, from $\left|Q_{h+1}^{(i)}(s,a) - \bar{\phi}(s,a)^\top \bar{\boldsymbol{\theta}}_i\right| \leq \sqrt{k}\alpha'$ we know that

$$\left|z_{h,j}^{(i)}\left(Q_{h+1}^{(i)}\right)\left(s_{h,j}^{(i)}, a_{h,j}^{(i)}\right) - z_{h,j}^{(i)}\left(\bar{Q}_{h+1}^{(i)}\right)\left(s_{h,j}^{(i)}, a_{h,j}^{(i)}\right)\right| \tag{84}$$

$$= \left|\max_{a \in \mathcal{A}} Q_{h+1}^{(i)}\left(s_{h+1,j}^{(i)}, a\right) - \mathcal{T}_h^{(i)}\left(Q_{h+1}^{(i)}\right)\left(s_{h,j}^{(i)}, a_{h,j}^{(i)}\right) - \max_{a \in \mathcal{A}} \bar{Q}_{h+1}^{(i)}\left(s_{h+1,j}^{(i)}, a\right) + \mathcal{T}_h^{(i)}\left(\bar{Q}_{h+1}^{(i)}\right)\left(s_{h,j}^{(i)}, a_{h,j}^{(i)}\right)\right| \tag{85}$$

$$\leq \max_{a \in \mathcal{A}} \left|Q_{h+1}^{(i)}\left(s_{h+1,j}^{(i)}, a\right) - \bar{Q}_{h+1}^{(i)}\left(s_{h+1,j}^{(i)}, a\right)\right| + \left|\mathcal{T}_h^{(i)}\left(\bar{Q}_{h+1}^{(i)} - Q_{h+1}^{(i)}\right)\left(s_{h,j}^{(i)}, a_{h,j}^{(i)}\right)\right| \tag{86}$$

$$\leq 2\sqrt{k}\alpha' \tag{87}$$

hence we have

$$\sum_{i=1}^{M} \left\langle \boldsymbol{z}_{h,t}^{(i)} - \bar{\boldsymbol{z}}_{h,t}^{(i)}, \hat{f}_h^{(i)}(\boldsymbol{X}_{h,t}) - \dot{f}_h^{(i)}(\boldsymbol{X}_{h,t}) \right\rangle \tag{88}$$

$$\leq \sum_{i=1}^{M} \left\|\boldsymbol{z}_{h,t}^{(i)} - \bar{\boldsymbol{z}}_{h,t}^{(i)}\right\| \cdot \left\|\hat{f}_h^{(i)}(\boldsymbol{X}_{h,t}) - \dot{f}_h^{(i)}(\boldsymbol{X}_{h,t})\right\| \tag{89}$$

$$\leq 2\alpha'\sqrt{MTk} \cdot \left\|\hat{f}_{h,t} - \dot{f}_{h,t}\right\|_{2, E_t} \tag{90}$$

holds for arbitrary $\{Q_{h+1}^{(i)}\}$ at any level $h \in [H], t \in [T]$.

Adding (83) and (90), we finally finish the proof by setting $\alpha = \alpha' = \frac{1}{MTk}$

$$B_{h,1} = \sqrt{2Mk + \log(\mathcal{N}(\Phi, (kMT)^{-1}, \|\cdot\|_\infty)/\delta)} + 1 \tag{91}$$

$$B_{h,2} = 2\sqrt{MT + \log(2MT^2/\delta)} \tag{92}$$

$\square$

## C  Experiment Dissection and Discussion

In this section, we will take a closer view of the learning procedure and analyze the functionality of the UCB term in our algorithm. Usually, a reasonable UCB term should embrace several properties. *(i)* It should let confidence set $\mathcal{F}_t$ contain the real parameter with high probability. *(ii)* It should shrink at a reasonable speed to achieve low regret.

To check (i), we choose the model $\hat{f}_t$ at step $t = 200$ which is trained on insufficient data with only 2000 samples. We then sample 100 images from test set as unknown inputs $\mathcal{D} = \{(\boldsymbol{x}_i, y_i)\}_{i=1}^{100}$, where $\boldsymbol{x}_i$ is the digit image and $y_i$ is the corresponding target value. We inspect the relationship between the original prediction error $|\hat{f}_t(\boldsymbol{x}_i) - y_i|$ and the added bonus $b_i = \bar{f}_t(\boldsymbol{x}_i) - \hat{f}_t(\boldsymbol{x}_i)$ via finetuning on each input $\boldsymbol{x}_i \in \mathcal{D}$. The result is presented as scatter dots in Figure 2(a). We can clearly see that almost all the points lie above the line $y = x$, meaning that $b_i = \bar{f}_t(\boldsymbol{x}_i) - \hat{f}_t(\boldsymbol{x}_i) \geq |\hat{f}_t(\boldsymbol{x}_i) - y_i| \geq y_i - \hat{f}_t(\boldsymbol{x}_i)$ for any $i \in [100]$, which further indicates that $\bar{f}_t(\boldsymbol{x}_i) \geq y_i$. This validates that we can always find some $\bar{f} \in \mathcal{F}_t$ to give an optimistic estimation of the value for almost every $\boldsymbol{x}$. Moreover, we can observe an apparent correlated pattern between the test error and bonus, which implies that our

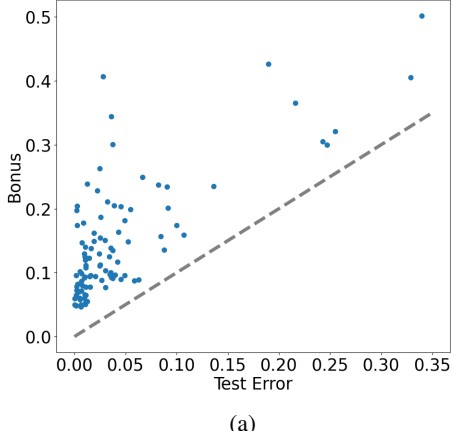
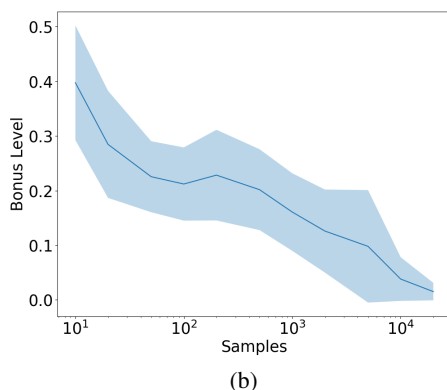

$$\text{(a)} \qquad\qquad\qquad\qquad\qquad\qquad \text{(b)}$$

Figure 1: (a) The relationship between unknown data's prediction error and the bonus it gets from finetuning. The grey line is $y = x$. (b) The average bonus level of 100 test images with respect to the number of samples in training set, the shaded area is the interval for $\pm 1$ standard deviation.

algorithm will give larger bonus for the data point whose prediction is not reliable, and only give relatively small bonus for the data that it is confident with.

We also check (ii) by plotting the average bonus level (closely related to the width of confidence set) against the number of samples the algorithm has been trained on. We gradually increase the number of samples from 10 to 20000 and fix a set of test images $\mathcal{D}$ as before to see how the average bonus level changes when the training set size increases. The result is shown in Figure 2(b). Previous work [**?**] proves that the eluder dimension of neural networks can be exponentially large in the worst case, which means that it can give almost arbitrary output value even when it is constrained to give a precisely accurate prediction for a large number of samples in the training set. In that case, the average bonus level should have remained constant regardless of the size of the training set. However, our experiment shows that the average bonus drops when the number of training samples increases. We conjecture that it is because in reality, when the input data are restricted to regular images with clear semantics, and the optimization procedure of the model is conducted via gradient-based methods in a very close neighborhood, the arbitrariness of the neural network's output is substantially reduced.

Restricting the model's training loss in the training set effectively limits the bonus obtained from the finetune procedure, which realizes the desired fast-shrinking property from our functional confidence set. Such a phenomenon sheds light on the unknown property of neural network's generalization capability and interpolation plasticity. We leave explaining the underlying mechanism as future work.

### C.1 Visualize the Learned Representation

A natural and interesting question is what representation does our CNN backbone actually learn. To investigate this problem and visualize the learned representation, we measure the information of different digits within the learned representation. Interestingly, we find that our model indeed learns an indicative representation for classification problem via multitask value regression training.

The basic measurement for the quality of representation is evaluated with the kernel function $\kappa(\boldsymbol{x}_i, \boldsymbol{x}_j) = \langle \phi(\boldsymbol{x}_i), \phi(\boldsymbol{x}_j) \rangle$ and see whether it has a strong diagonal. We take the checkpoint of neural network model at final step (around 600 with more than 6000 samples), and treat the module before the final linear layer as $\phi(\cdot)$. Denote the MNIST test set as $\mathcal{D} = \{\mathcal{D}_i\}_{i=0}^{9}$ where $\mathcal{D}_i$ is the images of digit $i$. Define the correlation between digit $i$ and $j$ under representation $\phi$ as

$$C(i, j) = \frac{1}{|\mathcal{D}_i| \times |\mathcal{D}_j|} \sum_{\boldsymbol{x}_s \in \mathcal{D}_i} \sum_{\boldsymbol{x}_t \in \mathcal{D}_j} \langle \phi(\boldsymbol{x}_s), \phi(\boldsymbol{x}_t) \rangle \tag{93}$$

To accelerate the evaluation, notice that we can preprocess an "template vector" $\boldsymbol{T}_i$ for each digit $i$ as

$$\boldsymbol{T}_i = \frac{1}{|\mathcal{D}_i|} \sum_{\boldsymbol{x} \in \mathcal{D}_i} \phi(\boldsymbol{x}) \tag{94}$$

so that the correlation can be computed through

$$C(i,j) = \frac{1}{|\mathcal{D}_i| \times |\mathcal{D}_j|} \sum_{\boldsymbol{x}_s \in \mathcal{D}_i} \sum_{\boldsymbol{x}_t \in \mathcal{D}_j} \langle \phi(\boldsymbol{x}_s), \phi(\boldsymbol{x}_t) \rangle \tag{95}$$

$$= \frac{1}{|\mathcal{D}_j|} \sum_{\boldsymbol{x}_t \in \mathcal{D}_j} \left( \frac{1}{|\mathcal{D}_i|} \sum_{\boldsymbol{x}_s \in \mathcal{D}_i} \langle \phi(\boldsymbol{x}_s), \phi(\boldsymbol{x}_t) \rangle \right) \tag{96}$$

$$= \frac{1}{|\mathcal{D}_j|} \sum_{\boldsymbol{x}_t \in \mathcal{D}_j} \left\langle \frac{1}{|\mathcal{D}_i|} \sum_{\boldsymbol{x}_s \in \mathcal{D}_i} \phi(\boldsymbol{x}_s), \phi(\boldsymbol{x}_t) \right\rangle \tag{97}$$

$$= \frac{1}{|\mathcal{D}_j|} \sum_{\boldsymbol{x}_t \in \mathcal{D}_j} \langle \boldsymbol{T}_i, \phi(\boldsymbol{x}_t) \rangle \tag{98}$$

$$= \langle \boldsymbol{T}_i, \boldsymbol{T}_j \rangle \tag{99}$$

We plot this 10x10 correlation map for single task training and multitask training with $M = 10$. Notice that the single task reward mapping function is $\sigma(i) = i/10$, and to assure the different tasks in multitask training are heterogeneous, we manually set that the best digit for each task are distinct.

The result is in figure 3. We can see that since single task only needs to recognize the large value digit, namely 9, 8 or 7, its representation function is not informative for distinguishing digits. And interestingly, the multitask trained network's representation demonstrates a very strong diagonal, indicating that the representation vector is very specific to the digit's image, although the training process has no explicit definition for the classification task but a regression problem instead. Actually, we found a simple linear layer append to this representation can achieve over 95% accuracy on MNIST test set.

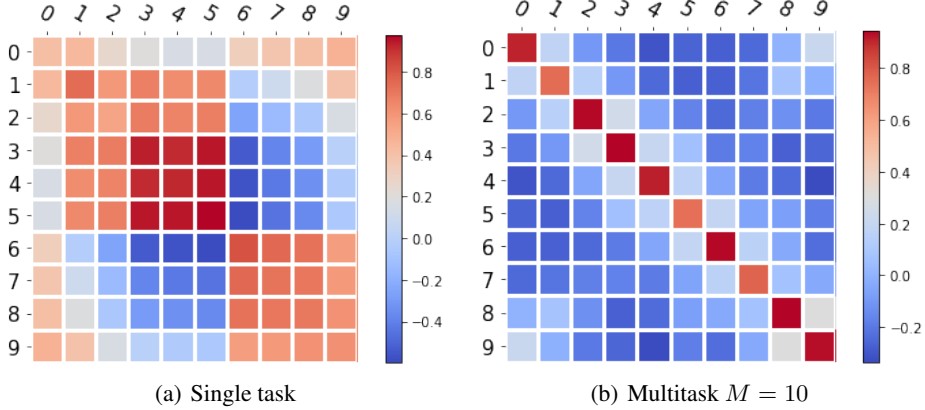

(a) Single task  (b) Multitask $M = 10$

Figure 2: The kernel function for the representation learned by single task and 10-tasks multitask. It is clear that multitask representation learning obtains a more comprehensive and interpretable pattern for the MNIST images.