# OpenReview forum: "Provable General Function Class Representation Learning in Multitask Bandits and MDP"
_NeurIPS.cc/2022/Conference — NeurIPS 2022 Accept_

### Official Review · Reviewer_1Kyf · 2022-07-11

**Rating:** 6
**Confidence:** 4
**Soundness:** 3 good
**Presentation:** 4 excellent
**Contribution:** 2 fair

**Summary:**

The paper proves the first near-optimal regret analysis for multitask bandits and MDPs with a general function class representation learning. Different from previous literature, this paper consider general representation function that is not known to the algorithm. The algorithms follow the well-known optimism-in-face-of-uncertainty design. The effectiveness of the algorithm for bandits is validated on MINIST dataset.

**Questions:**

Line 70: it would be better to point it out that this result is for contextual bandit setting.
Assumption 2.1: there is a typo.
Line 280: How is the performance boosted by $Md$? Should it be $\sqrt{Md}$?

**Limitations:**

I believe the main limitation of the proposed algorithms is that they are not practical. Discussing a practical algorithm for MDP case would be interesting.

**Strengths And Weaknesses:**

Strengths:

The paper adds to the line of works that try to understand the benefits of multitask representation learning on sequential decision-making problems. As far as I know, it is the first paper to consider the unknown representation function beyond linear class. The paper is well-written and easy to follow. The proofs are complete.

Weakness:

My concern is that the contribution of this paper is highly incremental. Benefits of the similar form have been extensively discussed in the literature though it is a relatively new setting in this paper. I briefly went through the proofs. The paper makes very limited technical contributions. If there is any significant technical contribution, I suggest the authors point it out in the main text. The paper discussed both bandit and MDP settings, while the experiments are only conducted for bandits problems. It would be good to discuss the practice algorithm for MDP as well.

It would be interesting to discuss the worst-case generalization performance for a learned representation, which seems to be a more interesting problem for multitask RL.

---

> ### Author Response · Authors · 2022-08-01
> **Response to Reviewer 1Kyf**
>
> Thank you for your time and valuable suggestions. Please see our response below.
> ## 1. Incremental contribution.
> In response to the concern of contributions, our work has the following novelties which overcome non-trivial obstacles that make previous works([14] and [25]) cannot apply.
>
>
> * **Not Rely on Linearity**. The regret analysis in [14] has a strong dependency on linearity. For instance, equation (88) in [14] at page 24 decomposes the regret into the product of input norm and parameter difference's norm, then bounds each of them. However, this is only possible for linear cases. Namely, only when $f_{\theta}(x)=x^{\top}\theta$ can we just simply constrain $\|\theta-\hat{\theta}\|$ to bound the error of *any* input's prediction by $|f_{\theta}(x)-f_{\hat{\theta}}(x)|=|x^{\top}(\theta-\hat{\theta})|\leq \|x\|\cdot \|\theta-\hat{\theta}\|$ for empirical risk minimizer $\hat{\theta}$. This trick solves the generalization issue and is probably the reason why most previous works only deal with linear cases. For the general function class, this is not applicable. The analysis has to bound the $\ell_{\infty}$-error $\max_{x\in X}|f_{\theta}(x)-f_{\hat{\theta}}(x)|$ where $X$ is the input space, and all we know about $f_{\hat{\theta}}$ is it minimizes the error for finite training set data. Achieving this requires a strong generalization guarantee and is the reason why we use Eluder Dimension (Lemma 0&2).
>
> * **Multi-head Function Class.** However, directly applying eluder dimension's [25] framework in multitask learning also faces other non-trivial technical obstacles. The efficacy of multitask representation learning essentially originates from the shared representation among tasks. It does not make any sense that concurrently solving $M$ totally non-related tasks can result in sublinear regret of $M$. Hence it is vital and necessary to characterize such relation between multiples tasks (functions) that the agent simultaneously learns. To this end, **we are the first to adopt a novel concept called multihead function class**, namely $\mathcal{F}^{\otimes M}$ in line 203. This abstract function space captures the essense for the relation between different tasks. Being more compact by sharing a common backbone $\phi$, it requires much less samples to learn, because all the tasks contirbute to shape a good representation $\phi$, then feedback to each task for faster convergence. This is the key point why multitask learning can reduce sample complexity, and is firstly revealed in our work. Its benefit is reflected in Lemma 1, where we utilize the particularity of $\mathcal{F}^{\otimes M}$ to achieve a much tighter bound for confidence set's width (depicted by the order of $\beta_t$), and further results in sharper regret bound. This is also absent in previous works.
>
> * **Empirics.** To the best of our knowledge, this is the first work that experimentally validates multitask non-linear representation learning's efficacy. Also, we firstly identify an intriguing phenomenon that empirical eluder dimension is much smaller than the worst exponential bound. This is a surprising result contrary to the common belief that neural networks are very good at memorizing particular outlying points with seemingly infinite interpolation capacity.
>
> Thank you for your suggestion, we will explicit emphasize these technical contributions in table or bullet list in future versions.
>
> ## 2. Worst case analysis.
> Yes, generalization in worst case is a very interesting question to discuss. In fact,
> previous work [10] has already shown that, even with slightest non-linearity, there exist
>  hard bandit instance that **any** algorithm needs exponential number of samples.
> Consider $f(x)=ReLU(\langle x,a \rangle - 1 + \epsilon)$ where $a$ and $x$ are both unit norm vectors, and $a$ is unknown to the agent. This function only has positive reward near $a$ with all zero reward values for the majority of the rest space. Unless the algorithm exhaustively searches into the neighborhood of $a$, all the reward value would be 0, giving no information on the bandit function. So *any* algorithm has to essentially go through every corner of the d-dimensional sphere to search for a, which requires exponential samples of $d$ and $1/\epsilon$.
>
> Therefore, the following works have shifted the focus on more regularized setting with additional assumptions rather than solving the worst case, since no algorithm can escape exponential dependency on dimensionality $d$ in the most general setting.
> ## 3. Sample efficiency boost.
> It should be $Md$. The performance measure for bandit algorithm is comparing the decreasing order of average regret w.r.t. total steps $T$. In this sense, suppose baseline use $T'$ steps to reach the same running avg regret as ours in step $T$, consider only the dominating term in both bounds, this gives
>
> $$ MHd\sqrt{T' \mathcal{N}(\Phi) } / T' = H\sqrt{MTd \mathcal{N}(\Phi)} / T $$
> which yields $T'=Md T$.

---

> > ### Comment · Reviewer_1Kyf · 2022-08-05
> > **Thanks for the responses**
> >
> > I have read the responses from the authors. I can see the contributions to the literature and I decide to raise my score to 6.

---

> > > ### Author Response · Authors · 2022-08-08
> > > **Thank you for your time**
> > >
> > > Thank you for your appreciation and valuable suggestions! It is very helpful for us to improve our work.

---

### Official Review · Reviewer_rCEX · 2022-07-11

**Rating:** 7
**Confidence:** 2
**Soundness:** 4 excellent
**Presentation:** 3 good
**Contribution:** 3 good

**Summary:**

The paper extends regret analysis for multi-task linear MDPs and bandit problems when a shared representation that is unknown to the agent is used across tasks. A general representation function class $\Phi$ and value function approximation class $\mathcal{L} \circ \Phi$ that is linear in $\Phi$ is used. To enable analysis in this setting, the eluder dimension of Russo et al. (2013) is adopted and $\mathcal{L} \circ \Phi$ is assumed to have bounded eluder dimension. In this setting, the paper provides novel regret bounds for a UCB algorithm for multi-task bandits and linear MDPs. In the bandit case for linear representation class, the paper states that the bound recovers the best known bound as a special case. For linear MDPs, the bound extends known bounds for the single task setting and achieves improved sample complexity over a naive baseline that does not share representations across tasks. An empirical analysis is provided as a sanity check to demonstrate reduced cumulative regret over time with increasing number of tasks.

**Questions:**

Minor suggestions (typos, etc.):
- L318: "boarder" should be "border"
- Maybe capitalize section headers 3.2 and 3.3.


**Limitations:**

I was unable to find discussion of limitations of the algorithm/analysis in the text.

**Strengths And Weaknesses:**

Strengths:
- To the best of my knowledge, the paper is the first to characterize how sample complexity is improved in the multi-task MDP setting when shared structure is assumed for MDPs with low inherent Bellman error (IBE).
- The fact that the previous known bound of [14] is recovered as a special case for multi-task bandits is convincing regarding the sharpness of the bounds.

Weaknesses:
- Low IBE (Assumption 2.1) for all tasks strikes me as a potentially strong assumption. It would have been interesting to see how the algorithm behaves when a small subset of tasks do not have low IBE. How robust would the algorithm be in that case? If cumulative regret indeed grows linearly in the worst-case IBE as predicted by Thm. 2, this might be a shortcoming of the algorithm.
- The mathematical extension of regret bounds from single task to the multi-task setting seems somewhat straightforward. Nevertheless, shared representation structure is a key differentiator.

---

> ### Author Response · Authors · 2022-08-01
> **Response to Reviewer rCEX**
>
> Thank you for your time and valuable suggestions. Please see the response below.
>
> ## 1. Is low IBE a too strong assumption?
> Low IBE, and its special case called linear MDP, is a very popular and widely used theoretical modeling in studying the role of representation learning in bandits and MDPs ([14], [15], [20], etc.). When the function class $\Phi$ has adequate expressivity, i.e. neural networks. There always exists certain assignment of the parameters to fit the transition dynamics almost perfectly, and this indicates a very low IBE. (see [15]) Therefore, low IBE is not an overly strong assumption, which embraces rich classes of real MDP problems [a].
>
> ## 2. Linear dependency on IBE.
> The essence of low IBE assumption is exactly that $\mathcal{I}$ is sufficiently small, usually negligible in magnitude compared with other terms, so that any polynomial dependency (even linear) on $\mathcal{I}$ or T is too small to consider.
>
> ## 3. Mathematical extension.
> Although seems straightforward, the theoretical modeling to characterize the relation between all the tasks which is the shared representation, is critically important. Without this shared structure, it does not make any sense why dealing with multiple tasks is more efficient than separately solving each single task. Therefore we propose an important and novel technique to analyze in function space $\mathcal{F}^{\otimes M}$ to capture it. It characterizes the essence that samples from all the tasks contribute to accelerate the convergence for the shared $\phi$. So it is more compact to learn compared with separately solve each individual task (line 281).
>
> ## References
> [a] Bellman Eluder Dimension: New Rich Classes of RL Problems, and Sample-Efficient Algorithms. Jin et al. *NeurIPS 2021*

---

> > ### Comment · Reviewer_rCEX · 2022-08-08
> > **Thank you for your responses.**
> >
> > I am not sure if I agree with the comment that "There always exists certain assignment of the parameters to fit the transition dynamics almost perfectly, and this indicates a very low IBE." With increasingly complex dynamics, I believe we need an increasingly rich function class to be able to assume low IBE. Regardless, I understand that the low IBE assumption is somewhat standard in this line of work. However, my major complaint was not that the paper assumed low IBE, but that it assumed low IBE *for all tasks* and that the bound scales with the worst-case IBE across all tasks. I would have felt more comfortable with a bound that scales as expected IBE under some task distribution (e.g. simple mean), so that if there are a few outlier tasks for which the function class isn't rich enough to capture their transition dynamics, shared representation is still beneficial. Still, I don't think this takes away much from the main contribution of the work.
> >
> > After reading the other reviews and authors' responses, I am raising my *confidence* score by 1.

---

> > > ### Author Response · Authors · 2022-08-08
> > > **Thank you for your suggestions**
> > >
> > > Your concern that low IBE holds for all tasks is too strong is a very novel and interesting question for this line of research. Whether the situation will become fundamentally different when this assumption changes into expected low IBE is a blank area in the literature. To the best of our knowledge, there is no paper discussing this new situation. But a simple bound can be directly deduced from the expected IBE to the largest IBE. Suppose we have that the expected IBE of $M$ tasks as
> > > $$\mathcal{I}^{exp}=\frac{1}{M} \sum_{k=1}^M \mathcal{I}_k$$, we have that
> > > $$\mathcal{I}^{mul}=\max_k \mathcal{I}_k \leq M \mathcal{I}^{exp}$$
> > > since all $\mathcal{I}_k \geq 0$. Hence it seems that the result will just deviate by a factor of number of tasks $M$.

---

### Official Review · Reviewer_8wSD · 2022-07-21

**Rating:** 3
**Confidence:** 3
**Soundness:** 2 fair
**Presentation:** 2 fair
**Contribution:** 2 fair

**Summary:**

The authors consider multitasks reinforcement learning where generalized representations function among the tasks and task-specific parameters will be estimated sequentially in order to minimize cumulative regret. Previous works only focus on either linear function class or single-task which limits their applicability. Inspired by them, the authors employ eluder dimension to handle the concept of general function class and treat the problem as linear MDP (as well as linear contextual bandit). In the end, they provide proof-of-concept experiments to evaluate the benefit of shared representation among tasks

**Questions:**

Firstly, I suggest the authors put a table to highlight the distinction from a single task to multiple tasks as well as from linear class to general class. As I mentioned in Weakness 1, I don't see the significant contribution made by this work. If the authors can convince me that challenges from previous works to this one are nontrivial, I will consider increasing my score.

Secondly, I hope the author can connect the experiment to your theorems. For example, N(\Phi, \alpha) apparently is the main feature distinct from [14], but how can I see its influence shown in the experiment? Can the authors please point out in the rebuttal?

**Limitations:**

I don't see any potential negative social impact from their work and the authors address limitations properly.

**Strengths And Weaknesses:**

Strength:
This work extends the results in [14] from linear representation class to the general class with the tools provided from [25]. Their upper bound (Thm 1 and 2) explain the cost for learning representations function, O(\sqrt{MdTlog N(\Phi)}), will be sublinear in M, hence this huge cost (compared to learning \theta_i for each task) will be mitigated by jointly learning multiple tasks.

Weakness:
1. It is a not surprising combination of previous works, where most relevant techniques can be found in [14] and [25]. i.e., [14] designed and analysised  the same style algorithms for learning linear representation for multitasks in linear bandits and RL; [25] (and [31]) demostrated the construction of confidence bound for general class. Although I have quickly checked the proofs and believe the correcness of the main theorems (there are many inequalities and stataments that I have to guess the reason behind them, hope the authors can improve their writing), it is unclear what this work contributes.

2. Provable but intractable. The main difference between this work and [14] should be the representation extractor and confidence set (Line 202, 208 and 262 in this work and definition 1 in [14]). As mentioned by the authors (line 39), it is relatively easy to construct in linear class but hard in general. I expected the authors show how they construct the confidence set (*), but it is disappointing they used CNN as the representation extractor rather than solving line 202 and 262. It is mainly because the algorithms presented in this work suffer intractable computational cost, hence this makes us hard to derive any useful insight from their theorems to the practice.

---

> ### Author Response · Authors · 2022-08-01
> **Response to Reviewer 8wSD (part 2/2)**
>
>
> ## 2. Intractability.
>
> Yes, rigorously solving the confidence set and selecting action within it is intractable. However, this form is just for the convenience of analyzing its power in reducing sample complexity. We emphasize that, when the focus is sample complexity, many works in theoretical reinforcement learning literature also simply assume an oracle to solve problems and the oracle is intractable (see [d],[e],[f]).
>
> In practice, it is not necessary to exactly solve all the candidate functions to explicitly construct confidence set, we can use graident methods to navigate within the set to search for good approximation solutions. Also, it is not necessary to rigorously compute the UCB that involves N(\Phi). All we need is a numeric bound ($\beta_t$ in (\*)) to constrain the total deviation of prediction in training set. Hence we simply tune it in form a*log(bt+c) and empirically showed it suffices to produce good results.
>
> In conclusion, the apparent difference in $\mathcal{N}(\Phi,\alpha)$ is just for convergence and correctness analysis. And it is just an upper bound for the UCB level that could work. In reality, there is no need for any algorithm to rigorously follow the formulation that involves intractable terms because it is not the essence of why it works. The critical point is that, by bounding the total prediction error on the training set as a regularization, the model-based algorithm can achieve better results through proper implicit exploration. Such procedure can also be seen in empirical RL (like BRAC in offline RL [g]).
>
> ## References
> [a] Improved Algorithms for Linear Stochastic Bandits, Yadori et al. *NeurIPS 2011*
>
> [b] Nearly Minimax Optimal Reinforcement Learning for Linear Mixture MDP. Zhou et al. *ICML 2022*
>
> [c] Bandit Algorithms, Csaba Szepesvari.
>
> [d] Bellman Eluder Dimension: New Rich Classes of RL Problems, and Sample-Efficient Algorithms. Jin et al. *NeurIPS 2021*
>
> [e] Contextual Decision Processes with Low Bellman Rank are PAC-Learnable. Jiang et al. *ICML 2017*
>
> [f] Model-based RL in Contextual Decision Processes. Sun et al. *COLT 2019*
>
> [g] Behavior Regularized Offline Reinforcement Learning. Wu et al.

---

> > ### Comment · Reviewer_8wSD · 2022-08-06
> > **Thank you for resonse**
> >
> > After reading the author's reply thoroughly, I decide to keep my score unchanged. The main reason is that the key step of their algorithm, computing F_t, is computationally intractable, hence the sublinear regret enjoyed by the algorithm is basically meaningless (which is the main contribution of this work). Though I agree that is not necessary to solve exactly, the authors cannot even provide any error bound for their surrogate (namely, CNN in experiment). For this reason, I think the algorithm provided in experiment should be viewed as an "another" algorithm rather than the guaranteed one. Also, speaking of reality, the experiment considered (MNIST) is far from the real problems. And even for this toy example, the guaranteed algorithm suffers unavoidable intractablilty, there is no hope for it in reality. In conclusion, I think the entire contribution is too less to be accepted.

---

> > > ### Author Response · Authors · 2022-08-07
> > > **Further Explanation about the Literature**
> > >
> > > We appreciate your requirement for the rigor in committing theoretical algorithms into practice, and understand your concerns in its intractability and inefficiency. However, your concerns are applicable to a long line of  papers in the RL and bandit, including some influential and seminal works such as [25] and [31]. In particular, these works rely on optimization oracles for solving NP-hard problems just like many other works [d][e][f][h], where the policy class from a neural network has intractable cardinality in general. Therefore, it is impossible to derive rigorous bounds (unless P = NP) for arbitrary super complex modern neural networks. But it does not diminish these works' contributions. Some papers, including ours, use (heuristic) approximation algorithms to implement the optimization oracle. Furthermore, some works rely on optimization oracles which are even difficult to implement a heuristic approximation, and thus they were unable to conduct experiments [14][e][31]. And our approximation has demonstrated its effectiveness in the experiment section, thus is not "unavoidable intractable". Its approximation version paradigm (like [g] we mentioned) is ubiquitous in real practice, our work can also be viewed as an explanation for their efficacy.
> > >
> > > Lastly, we would like to note again that these works are **not** meaningless just because of these intractabilities, since they still provide useful (theoretical / practical )insights. Providing sample complexity guarantee is significant enough in contributions from theoretical perspective, just like many seminal and influential works in the field before.
> > >
> > > [h] Taming the Monster: A Fast and Simple Algorithm for Contextual Bandits. Agarwal et al. *ICML 2014*

---

> > > > ### Comment · Reviewer_8wSD · 2022-08-07
> > > > **Explanation on my comment**
> > > >
> > > > Thank you for the fast reply to my comment. Though I am concerned about the intractable issue in this work, this comment should not be directly extended to the influential works given above. They have valuable insight which shed light on the follow-up works even if some steps are intractable. In the other words, intractability is indeed their drawback, which should not be overlooked, but their positive impact is significant.
> > > >
> > > > However, I am afraid that your work is not such a case. Given that the linear class for multitasks in RL has been well-studied [14] and extension from linear class to general function class is verified [25], the theoretical contribution of this work, an "intractable" algorithm for general class in multitasks, is not as insightful as the previous works.

---

> > > > > ### Author Response · Authors · 2022-08-07
> > > > > **Revist contribution and insight issue**
> > > > >
> > > > >
> > > > > ## Insight and Contributions.
> > > > > In our initial response to your review, we have listed the fundamental novelties and contributions that our work differs from previous works in order to address your concern on "unclear about what this work contributes".
> > > > >
> > > > > Although it appears that we just use non-linear tools in [25] and plug it into [14]'s linear analysis framework, this is actually not true. **Nontrivial obstacles need to be solved when combining these two methods, and its solution reveals the essence for the benefit of multitask representation learning in bandits and MDPs in general cases.** It is not a simple A+B work that trivially combines two techniques from past works, but requires fundamental novel formulation of the function space to achieve low regret bound. The analyzing framework need to characterize the relation between multiples tasks (functions) that the agent simultaneously learns, since it makes no sense that solving $M$ totally unrelated tasks can result in any benefit. So we firstly proposed the concept of multihead function $\mathcal{F}^{\otimes M}$ that captures it. Namely, separately solving each task is learning in space
> > > > > $$\mathcal{F}^M=\{(f^{(1)},f^{(2)},...,f^{(M)}):f^{(i)}(\cdot)=\phi_{i}(\cdot)^{\top} w_i,i=1,2,...,M \}$$
> > > > > while ours is learning in space
> > > > > $$\mathcal{F}^{\otimes M}=\{(f^{(1)},f^{(2)},...,f^{(M)}):f^{(i)}(\cdot)=\phi(\cdot)^{\top} w_i,i=1,2,...,M \}$$
> > > > > The main difference is that now all the functions $f^{(i)}$ share a common backbone $\phi$ rather than individual $\phi_i$, hence the complexity (also can be vaguely understood as degrees of freedom) of the function class $\mathcal{F}^{\otimes M}$ is much smaller than $\mathcal{F}^M$. Utilizing this particular structure of $\mathcal{F}^{\otimes}$, we achieve a much tighter bound for the confidence set width $\beta_t$. Intuitively speaking, in multitask learning, all the tasks contribute to shape a good representation $\phi$, then feedback to each task to learn each task-specific parameter $w_i$ to achieve faster convergence and lower regret. This is the key point why multitask learning can reduce sample complexity, and is firstly revealed in our work.
> > > > >
> > > > > We will sincerely appreciate it if you could tell us in detail what are exactly the reasons that our work lacks useful insights and contributions compared with previous works. It would be much more helpful for us to improve our work if you could explain it with constructive suggestions. Thank you very much.

---

> > > > > > ### Comment · Reviewer_8wSD · 2022-08-08
> > > > > > **Regarding the author's question**
> > > > > >
> > > > > > My comment cannot be directly applied to the previous works since the main contributions of this work and the previous fundamentally differ from each other. [25] introduce a new notion of measure, Eluder dimension, successfully capturing many desired merits for many types of function classes including linear function class, generalized linear class, Lipschitz function class, and so on. Though their algorithm involves an intractable oracle for the general function class, their main contribution (from my understanding) is to demonstrate how Eluder dimension affects the standard algorithms such as UCB and Thompson Sampling in a wide class of frameworks. However, the contribution of this submitted work apparently is NOT to introduce any new measures. Directly applying my concern about the intractability to [25] is overly twisted.
> > > > > >
> > > > > > The author argued their contribution is introducing a multi-head function class and showing it combined with a UCB-style algorithm attain sublinear regret. But unfortunately, such sublinear regret is attainable for an "intractable" algorithm, and I think what people are interested in is whether a tractable algorithm can do it or not. One suggestion I can give (perhaps not so constructive, sorry) is to build a simplified function class that avoids intractability and relate eluder dimensions of the original and simplified one.
> > > > > >
> > > > > > In the end, I think this discussion has derailed. The goal of discussion in the author-reviewer period should be to figure out the misunderstanding parts in the submitted work rather than to discuss other works. After the intensive discussion, I promise that I will reassess this work, please wait patiently. Thanks.

---

> > > > > > > ### Author Response · Authors · 2022-08-08
> > > > > > > **Thank you for your suggestion and more explanation on potential misunderstandings**
> > > > > > >
> > > > > > > ## Revised version submitted
> > > > > > > Thank you for your time and constructive suggestions. We agree that the discussion should back on track. We have carefully read your suggestion and agree with your opinion that a more tractable algorithm would be much more practical and beneficial for reality. Your thoughtful suggestions has been considered and relevant discussion has been added in our revised version regarding intractability and its practical usage (line 231-256 in supplementary) to address your concerns. During your reassessment, it may be useful to give some further explanations to eliminate potential misunderstandings, and this may help you better understand the contribution of our work.
> > > > > > > ## Is the algorithm in our experiment totally another algorithm?
> > > > > > >  According to our understanding, the answer is no. The main difference between our practical version algorithm and the theoretical one is that we did *not* list out all the functions in the *whole* confidence set explicitly, but just use gradient-based method to search within a very small fraction of them with heuristics. Getting a (some) candidate within the confidence set is much easier (and certainly tractable) than solving **all** of them since we already have empirical minimizer $\hat{f}_t$ as a candidate and we can just search in the neighborhood around it. Rigorously speaking, our tuned numeric $\beta_t$ is much smaller than the theoretical guaranteed ones, so all the candidate functions $\tilde{f}_t$ that we search for still satisfy the theoretical requirement (but may omit some). Therefore, our practical version algorithm should be regarded as an inaccurate approximation to the theoretical algorithm, but not totally another algorithm. **A coarse approximation could already result in a successful boost of efficiency** is actually the evidence for the effectiveness of our algorithm.
> > > > > > > ## Can we derive any useful insight from this work?
> > > > > > > From our perspective, the answer is yes. There are at least three novel messages that our work conveys to the community. **And the main contribution of this work is much more than just giving a sub-linear regret bound**. This is an interpretation-oriented work rather than a pragmatic one.
> > > > > > > * **What is the essence of multitask representation learning helping reduce sample complexity in general case?** Our formulation of multihead function space $\mathcal{F}^{\otimes M}$ answers this question by revealing the critical procedure that multitask learning utilize samples from all tasks to locate the common backbone (or representation extractor) $\phi$, and this accelerates the convergence of $\phi$, which in turn boost the sample efficiency for each task and result in lower regret.
> > > > > > > * **Why does offline RL benefit from properly regularizing the policy's prediction error on offline dataset?** The concept of "behavior-regularization", which adds a penalty term to the model's prediction error on offline dataset, can be regarded as a practical version for constructing a *confidence set* and searching within it. This coincides with the procedure that approximately optimizes objective in line 311, where the model has to optimize online reward and keep its prediction error on training set to be small. This implies that the successful practice of behavior regularization algorithm (BRAC) actually benefits from implicit exploration within this confidence set. Our work provides with a brand-new perspective for understanding its effectiveness.
> > > > > > > * **Under some regularization conditions, the interpolation capacity of neural networks (like CNN) greatly decreases.** The worst-case bound for eluder dimension $d$ of a non-linear model like neural networks is exponential in its number of parameters, hence some (including us at the first time) may concern that this makes the bound become meaningless. However, in reality, we found that when the data follows some regular distribution (like digit images or natural images) and the selection of model parameters is influenced by the inductive bias of gradient-based methods, the empirical eluder dimension is much smaller and makes the algorithm tractable (see our discussion in Appendix, where the bonus level shrinks). This is a surprising result even for the machine learning community beyond this paper's scope.
> > > > > > > ## A simplified function class that avoids intractability.
> > > > > > > This is actually contained within our work. When $\Phi$ is linear, the confidence set $\mathcal{F}_t$ becomes an ellipsoid, and its update is definitely tractable via covariance matrix. We have shown in line 239 of our paper that, in this case, our algorithm reduces to near-optimal policy for linear bandits with shared representations. As for more non-linear but tractable function classes with analytical solutions,  please refer to [I]. We hope this can address your concern on intractability to some extent.
> > > > > > > ## Reference
> > > > > > > [i] Understanding the Eluder Dimension, Li et al.

---

> ### Author Response · Authors · 2022-08-01
> **Response to Reviewer 8wSD (part 1/2)**
>
> Thank you for your time and valuable suggestions. We appreciate your advice in improving the writing and will add more explanations on detailed deductions and intuitions. Please see the response below.
>
> ## 1. Contribution
> As the current standard paradigm for bandit problems, all algorithms (such as [14], [15], [b]) adopting OFUL share the highly similar style procedure and proof techniques in general (figure(1) in [a], sec 19.3 in [c]). Therefore, the inevitable resemblance of algorithm procedure and analysis does not necessarily mean the lack of contribution. The difference mostly lies in UCB form and function space to be learned, where the differences and non-triviality are more subtle.
>
> Below we summarize what are the non-trivial obstacles that make previous works([14] and [25]) cannot apply in our setting, and how we overcame them. Although it appears that the overall algorithm and techniques are similar, our work still embraces substantial novelties and contributions.
>
> * **New Setting**. Under same paradigm of OFUL algorithms, extensions in different settings i.e. multitask representation learning is critical. To the best of our knowledge, this the first *general* function class representation learning analysis in multitask bandits and MDPs. It is a substantial step forward in understanding multitask representation learning, which completes the blank in current literature.
>
> * **Not Rely on Linearity**. The regret analysis in [14] has a strong dependency on linearity. For instance, equation (88) in [14] at page 24 decompose the regret into the product of input norm and parameter difference's norm, then bounds each of them. However, this is only possible for linear cases. Namely, only when $f_{\theta}(x)=x^{\top}\theta$ can we just simply constrain $\|\theta-\hat{\theta}\|$ via low empirical risk to bound the error of *any* input's prediction by $|f_{\theta}(x)-f_{\hat{\theta}}(x)|=|x^{\top}(\theta-\hat{\theta})|\leq \|x\|\cdot \|\theta-\hat{\theta}\|$ for empirical risk minimizer $\hat{\theta}$. This trick solves the generalization issue, and is probably the reason why most previous works only deal with linear cases. For general function class, this is not applicable. The analysis has to bound the $\ell_{\infty}$-error $\max_{x\in X}|f_{\theta}(x)-f_{\hat{\theta}}(x)|$ where $X$ is the input space, and all we know about $f_{\hat{\theta}}$ is it minimizes the error for finite training set data. Achieving this requires a strong generalization guarantee and is the reason why we use Eluder Dimension (Lemma 0&2).
>
> * **Multi-head Function Class.** However, directly applying eluder dimension's [25] framework in multitask learning also faces other non-trivial technical obstacles. The efficacy of multitask representation learning essentially originates from the shared representation among tasks. It does not make any sense that concurrently solving $M$ totally non-related tasks can result in sublinear regret of $M$. Hence it is vital and necessary to characterize such relation between multiples tasks (functions) that the agent simultaneously learns. However, such structure is absent in previous single task work like [25] or [31], and it calls for special techniques to analyze the efficiency for learning these distinct-but-correlated functions. To this end, **we are the first to adopt a novel concept called multihead function class**, namely $\mathcal{F}^{\otimes M}$ in line 203. This abstract function space captures the essence for the relation between different tasks. Being more compact by sharing a common backbone $\phi$, it requires much fewer samples to learn, because all the tasks contribute to shape a good representation $\phi$, then feedback to each task for faster convergence. This is the key point why multitask learning can reduce sample complexity, and is firstly revealed in our work. Its benefit is reflected in Lemma 1, where we utilize the particularity of $\mathcal{F}^{\otimes M}$ to achieve a much tighter bound for confidence set's width (depicted by the order of $\beta_t$), and further results in sharper regret bound. This is also absent in previous works.
>
> * **Empirics.** To the best of our knowledge, this is the first work that experimentally validates multitask non-linear representation learning's efficacy. Also, we firstly identify an intriguing phenomenon that empirical eluder dimension is much smaller than the worst exponential bound. This is a surprising result contrary to the common belief that neural networks are very good at memorizing particular outlying points with seemingly infinite interpolation capacity.
>
> Thank you for the suggestion on highlighting distinctions, we will explicitly list them or make a table in future versions.

---

### Official Review · Reviewer_jNGD · 2022-07-22

**Rating:** 6
**Confidence:** 2
**Soundness:** 3 good
**Presentation:** 4 excellent
**Contribution:** 4 excellent

**Summary:**

This paper studies multitask representation learning with *general* function class in contextual bandits and MDP setting. In this setting, the goal is to learn a shared representation function that maps the state (or context) and action to a $k$-dimensional vector that allows the value function to be expressed as a linear function of this representation. They proposed an algorithm, Generalized Functional Upper Confidence Bound (GFUCB) which adapts UCB for a general function class in multitask setting and prove a regret bound for both the contextual bandits setting and multitask episodic RL setting with M related tasks (sharing common state and action space). Experiments were conducted in the non-linear neural network bandits setting based on the MNIST dataset to demonstrate that their GFUCB algorithm indeed outperforms naive epsilon greedy on a single task and scales to a larger number of tasks (i.e. lower cumulative regret with more tasks).

**Questions:**

1. While the focus is on the theoretical analysis in this paper, one possible improvement that can further strengthen the paper is providing also a proof of concept experiment for a linear MDP. Perhaps it can be built off the MNIST dataset as well (such as translating certain digits to their designated spot in each of the multitask instantiation), or a grid world fed as an image into a CNN.
2. A suggestion: it will be helpful if you summarize the practical approximations to Algorithm 1 in the Experiment section 6.2 as a separate “Algorithm 3” to show the correspondence between the original GFUCB algorithm and the practical instantiation (with running gradient descent for the optimization step, etc.).


**Limitations:**

The authors did not explicitly discuss the limitations of their work beyond stating the assumptions for the analysis. One observation noted in the appendix was that while the eluder dimensions of neural networks can be exponentially large, in practice (as shown in their experiment), the UCB bonus term still shrinks as the number of training samples increases. Thus specific interplay of using neural networks as the function class and its generalization / interpolation behavior is up for future study for this problem setting.

As a mainly theory paper, the authors specified “N/A” for potential negative societal impact. Perhaps we can still speculate on potential harmful uses of highly efficient multitask RL training algorithms leading to an agent capable of very large number of tasks.


**Strengths And Weaknesses:**

**Originality**: The paper presents the first provably sample efficient (with regret bound) algorithm for general function class representation learning in contextual bandits and linear MDPs. The authors clearly positioned their contributions in the background of prior works, which either studied multitask representation learning in linear representation functions or assumed prior known representation functions, or only studied single task setting when tackling general function class value approximation for bandits and MDPs.

**Quality**: The paper appears to be technically sound. The definitions and assumptions for the main results are given leading up to the algorithm and regret bound. The empirical experiments however only tackle the bandits setting, but not an MDP setting.

**Clarity**: While the paper is fairly dense in mathematical notations, overall the paper is adequately well-written. The intuitive interpretation of several theoretical results was helpful.

**Significance**: Given the abundant interest in multitask deep RL settings, this theoretical result is a step towards a deeper (pun intended) understanding in the community. The analysis in this paper and the resulting algorithm can give further inspiration for principled approaches in designing more practical algorithms with deep neural networks as well.

---

> ### Author Response · Authors · 2022-08-01
> **Response to Reviewer jNGD**
>
> Thank you for your time and valuable suggestions. Please see the response below.
>
> ## 1. About MDP experiments.
> Thank you for your detailed proposals about conducting MDP experiments. Due to the page limitation and intrinsic complexity for MDPs, we only conduct experiments for bandits. The main purpose of the experiment is to validate our framework's efficacy in extracting general non-linear representation in multitask learning. Therefore, bandit environment suffice to check this point. Still, to the best of our knowledge, this is the first work that empirically affirms the effectiveness of multitask representation learning in general function class in bandits.
>
> In our future work, we will take your advice and complete experiments in MDP.
>
> ## 2. Relationship between Algorithm 1 and practical experiments.
> Thank you for your advice. It should have been explained what are the relations between the approximation version experiment in section 6.2 and Algorithm 1. The main differences lie in two aspects:
> 1. We do not exactly compute UCB with complex net-covering number N(\phi). Instead, we simply tune the $\beta_t$ in (\*) in form a*log(bt+c) as a function of t.
> 2. We do not explicitly construct the confidence set. The gradient method implicitly explores within the confidence set to find good approximation solutions. In practice, we show that it suffices to produce good results.
>
> Apart from these approximations, our experiment carefully follows the procedure in algorithm 1 and corroborates our theoretical findings.
>
> ## 3. Limitations.
> There do exist some limitations in our work. Probably the main limitation is that our UCB bound is agnostic to data distribution and the structure of general function class $\Phi$. A simple but loose upper bound including eps-net covering number $\mathcal{N}(\phi)$ is used to deduce our results. Therefore, our framework fails to explain why empirical Eluder Dimension for a simple CNN on MNIST dataset is much smaller than the worst case bound. This is essentially an orthorgnal research topic that studies the generalization behavior of neural networks. And we hope future work can explain this. Probably, the data distribution together with the inductive bias that we select parameters via gradient method play a role of implicit regularization.

---

> > ### Comment · Reviewer_jNGD · 2022-08-07
> > **Thank you for your response**
> >
> > I have read the response from the authors, which addressed the difficulty of MDP experiments, clarified on the relationship between algorithm 1 and the practical experiments, and explained the limitations of the work. It will be good if these clarifications can be incorporated into the future version of the paper itself as well. I am keeping my score and confidence from my original review.

---

### Meta-Review · Area_Chair_TWzT · 2022-08-27

**Recommendation:** Accept
**Confidence:** Less certain

**Metareview:**

This paper investigates the problem of multitask representation learning with general function class in contextual bandits and MDPs. The assumption is that functions of interest (e.g. the average reward in the bandit setting) share, among various tasks, a common representation. For example, in contextual bandits, this representation corresponds to that used in contextual linear bandits but where the feature maps are the same for the various tasks. Here we need to learn this feature map along with the remaining parameters. Compared to previous works, see Hu et al., the possible feature maps are not linear but belong to some given functional space. The authors propose GFUCB, an algorithm that extend the idea of UCB to this setting (the confidence intervals of UCB are replaced by a subset of likely functions). The authors derive regret upper bounds for their algorithms. These bounds illustrate the gains achieved leveraging the information gathered across tasks.

The paper received 4 informative and insightful reviews. On the positive side, the paper is the first to give regret upper bounds for these kind of RL problems with representation learning. All reviewers further acknowledge the importance of this learning problems in practice.

On the less positive side, reviewers have raised some concerns and have suggested ways to improve the manuscript.
Among the concern, is the intractability of implementing the algorithm in practice. For example, the authors do not comment on how to compute $\hat{f}_t$ and (*) in practice. These quantities are at the core of the algorithm. We could say that an Oracle will solve these problems, but the authors could try to explain what kind of optimization problem we are facing and if there is a chance to solve it. This intractability limits the interest of the paper.
Some reviewers were also concerned about the novelty of the approach (combining techniques from [14] and [25]). The paper is judged incremental. I believe that the authors have done more than just combining techniques, but the paper is very unclear about the technical novelty. The authors do not explain any intuition behind the proofs. The latter are actually, in my view, extremely hard to follow and to check. The presentation has to be significantly improved before we are able to really assess the contributions. In their rebuttal, the authors try to explain the novelty of their approach; all these arguments should be clear and apparent in the paper.




**Award:**

No

---

### Decision · Program_Chairs · 2022-09-14

Accept